# A concerted ATPase cycle of the protein transporter AAA-ATPase Bcs1

Yangang Pan[1], Jingyu Zhan [2], Yining Jiang [1,3], Di Xia [2] & Simon Scheuring [1,4,5] ✉

Bcs1, a homo-heptameric transmembrane AAA-ATPase, facilitates folded Rieske iron-sulfur protein translocation across the inner mitochondrial membrane. Structures in different nucleotide states (ATPγS, ADP, apo) provided conformational snapshots, but the kinetics and structural transitions of the ATPase cycle remain elusive. Here, using high-speed atomic force microscopy (HS-AFM) and line scanning (HS-AFM-LS), we characterized single-molecule Bcs1 ATPase cycling. While the ATP conformation had ~5600 ms lifetime, independent of the ATP-concentration, the ADP/apo conformation lifetime was ATP-concentration dependent and reached ~320 ms at saturating ATP-concentration, giving a maximum turnover rate of $0.17 \, s^{-1}$. Importantly, Bcs1 ATPase cycle conformational changes occurred in concert. Furthermore, we propose that the transport mechanism involves opening the IMS gate through energetically costly straightening of the transmembrane helices, potentially driving rapid gate resealing. Overall, our results establish a concerted ATPase cycle mechanism in Bcs1, distinct from other AAA-ATPases that use a hand-over-hand mechanism.

Protein translocation across the cell membrane is a key process for cellular function[1–3]. Most proteins are translocated as unfolded polypeptide chains. However, the Rieske iron-sulfur protein (ISP) is known to translocate across the inner mitochondrial membrane (IMM) in the folded state. In eukaryotes, ISP is first translocated from the cytoplasm to the mitochondrial matrix in an unfolded state, is then folded in the matrix with the addition of the 2Fe-2S cluster to its C-terminal globular domain[4–6], from where it is, finally, as a matured and folded ISP translocated to the intermembrane space (IMS) and inserted into the mitochondrial complex III, *aka* cytochrome $bc_1$ complex (a ubiquinol-cytochrome c oxidoreductase)[7,8]. Within complex III, the ISP C-terminal 2Fe-2S cluster domain is exposed to the IMS, while the N-terminal transmembrane helix (TMH) is inserted into the IMM[9]. ISP translocation from the matrix to the IMS across the IMM is mediated by the transmembrane AAA-ATPase Bcs1[10–12]. Consequently, mutations in

Bcs1 lead to the deficiencies of complex III, which cause various severe mitochondrial diseases[13,14].

Bcs1 belongs to the superfamily of AAA proteins, ring-shaped homo oligomeric ATPases associated with numerous cellular activities including protein unfolding and degradation, DNA replication and membrane fusion[15–17]. Typically, soluble AAA proteins are hexamers and their structures revealed that the subunits arranged in a staircase-like configuration where all three nucleotide states, ATP, ADP and apo were found in the same complex. Based on these findings a chronological hand-over-hand mechanism of action was proposed[18–22]. In contrast, Bcs1 forms a homo-heptameric transmembrane protein, where all subunits are in the same conformation: Its large C-terminal domain, consisting of a Bcs1-specific domain and a nucleotide binding domain, forms a ~40 Å diameter and ~40 Å deep cavity facing the mitochondrial matrix, and a conical cavity in the transmembrane

[1]Department of Anesthesiology, Weill Cornell Medical College, New York, NY, USA. [2]Laboratory of Cell Biology, National Cancer Institute, National Institutes of Health, Bethesda, MD, USA. [3]Biochemistry & Structural Biology, Cell & Developmental Biology, and Molecular Biology (BCMB) Program, Weill Cornell Graduate School of Biomedical Sciences, New York, USA. [4]Department of Physiology & Biophysics, Weill Cornell Medical College, New York, NY, USA. [5]Kavli Institute at Cornell for Nanoscale Science, Cornell University, Ithaca, NY, USA. ✉e-mail: sis2019@med.cornell.edu

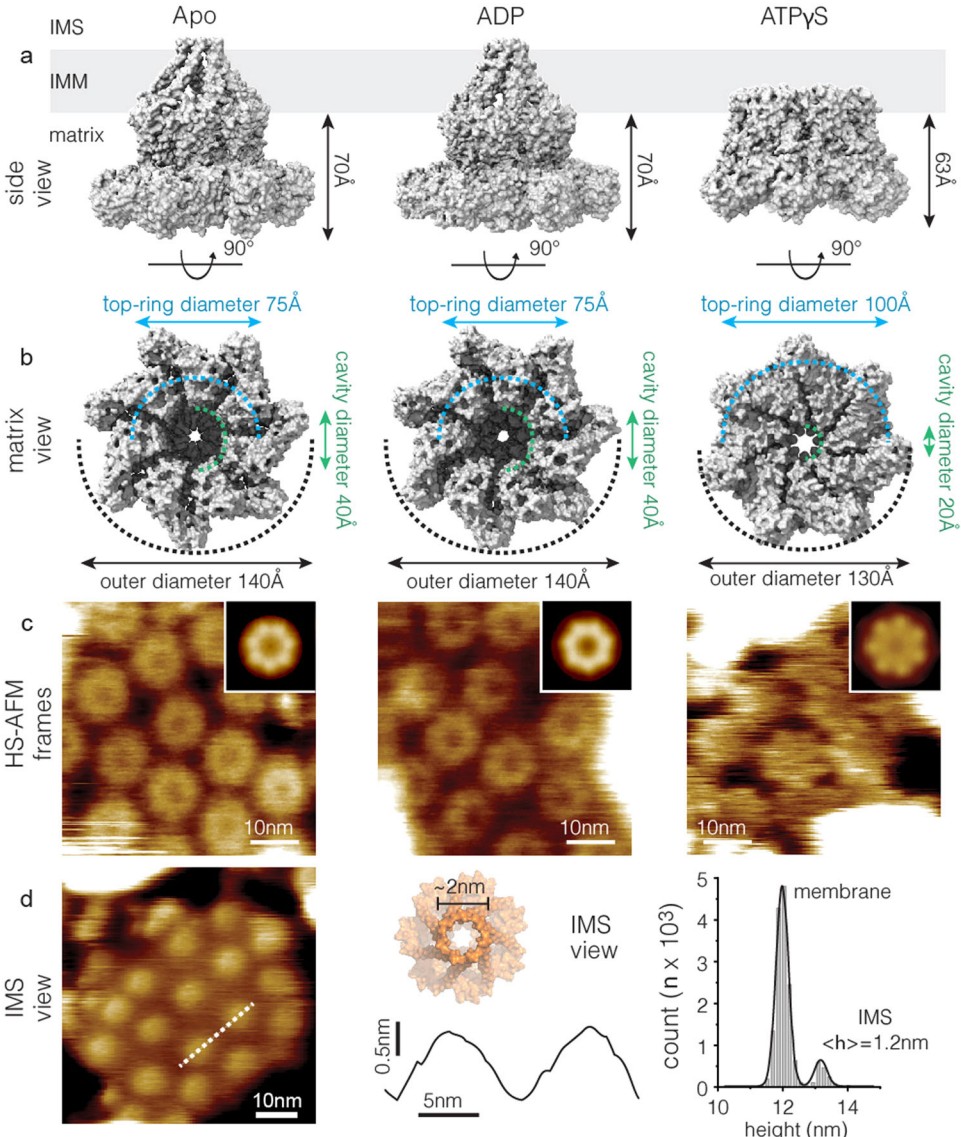

**Fig. 1 | HS-AFM analysis of Bcs1, sidedness and apo, ADP and ATPγS conformations. a** Side view of the apo (*left*), ADP (*middle*), and ATPγS (*right*) bound structures (PDB 6UKP, 6UKO, 6SH3, 6SH4, 6UKS) with TMD (only resolved for apo and ADP-bound states) and large C-terminal domain facing the matrix side. **b** Matrix side view of the structures as in **a**. The diameter of the most protruded parts of the seven Bcs1 subunits is ~75 Å for apo and ADP conformations (*left* and *middle*, RecA-like domain), and ~100 Å for ths ATPγS conformation (*right*, helical domain), respectively. **c** High-resolution HS-AFM images of apo (*left*), ADP (*middle*) ATPγS (*right*) bound Bcs1 viewed from matrix side. Insets: Correlation averages of the HS-AFM images of apo (*n* = 49, *left*), ADP (*n* = 67, *middle*), and ATPγS (*n* = 112, *right*) bound Bcs1. **d** Left: High-resolution HS-AFM image of apo Bcs1 viewed from the IMS side. Middle: Height profile along dashed line in left panel. Right: Height distribution analysis of left panel (*n* = 15,051 data points), with membrane, ~12 nm, and IMS protrusions, ~13.2 nm, above mica. The protrusion height of the Bcs1 IMS face, Δheight ~1.2 nm, above the membrane. Similar results as in **c**, **d** were observed in a minimum of 3 independent biological experiments (*n* = 3). Source data are provided as a source data file.

domain (TMD) in the apo conformation (Fig. 1a,b, left)[23,24]. The ADP-bound Bcs1 retained a very similar conformation as apo Bcs1 (Fig. 1a,b, middle). Instead, upon ATPγS (non-hydrolyzable ATP) binding, Bcs1 displayed a substantially different conformation, i.e., the matrix cavity size was reduced by nearly 70% and the height of the nucleotide binding domains below the membrane deceased by ~7 Å (Fig. 1a,b, right). These Bcs1 structures represented a tremendous progress as they allowed to propose a functional model where the folded ISP substrate protein is first loaded to the matrix cavity and then somehow, upon ATP-binding and -hydrolysis, pushed into the TMD cavity, from where it is released into the IMS. However, these static snapshots did not provide kinetic insights, nor could they constrain the functional cycle and its transitions and intermediates. In addition, in the apo/ADP conformation, all the 7 TMHs of Bcs1 were shown to

constitute an unusual basket-shaped structure that formed the TMD cavity[23,24] (Fig. 1a, left and middle), while the TMHs in the ATPγS-bound conformation remained unresolved. Overall, the working mechanism for Bcs1 remains poorly understood.

Here, we investigated the conformational changes of Bcs1 upon ATP-binding and -hydrolysis under close-to-physiological conditions, i.e., in membrane, in buffer solution and at ambient temperature and pressure, using high-speed atomic force microscopy (HS-AFM). HS-AFM has formerly been successful to study other ATPases, such as F₁-ATPase[25], V₁-ATPase[26], ClpB[27] and Abo1[28]. We found that the Bcs1 C-terminal domain facing the matrix side was in substantially different conformations in apo/ADP and ATPγS conditions with a ~7 Å protrusion height difference. In the presence of ATP, the domain switched back and forth between ATP-bound and apo/ADP-bound

**Table 1 | The sidedness of the Bcs1 membrane protruding surfaces in cryo-EM structures and HS-AFM**

| Technique | Feature | Matrix surface | IMS surface |
|---|---|---|---|
| Cryo-EM (apo) | outer diameter | 14.0 nm | — |
| | top-ring diameter | 7.5 nm | 2.0 nm |
| | protrusion height | ~7.0 nm | ~1.0 nm[a] |
| HS-AFM (apo) | center-to-center distance | ~15 nm | — |
| | top-ring diameter | ~7.0 nm | ~5.0 nm |
| | protrusion height | 6.9 nm | 1.2 nm |

[a]The protrusion height of the IMS surface is not clearly defined in the cryo-EM map but is estimated based on the hydrophobicity profile shown in[23].

**Table 2 | Structural features of the apo, ADP- and ATP- (ATPγS-) bound states Bcs1 matrix surface in cryo-EM structures and HS-AFM**

| Technique | Feature | apo | ADP | ATPγS |
|---|---|---|---|---|
| cryo-EM | outer diameter | 14.0 nm | 14.0 nm | 13.0 nm |
| | top-ring diameter | ~7.5 nm | ~7.5 nm | ~10 nm |
| | cavity diameter | 4.0 nm | 4.0 nm | 2.0 nm |
| | protrusion height | 7.0 nm | 7.0 nm | 6.3 nm |
| HS-AFM | center-to-center | ~15.0 nm | ~15.0 nm | ~14.0 nm |
| | top-ring diameter | ~7.0 nm | ~7.0 nm | ~10 nm |
| | cavity diameter | ~3.0 nm | ~3.5 nm | ~1.6 nm[a] |
| | protrusion height | ~6.9 nm | ~6.9 nm | ~6.2 nm |

[a]The measurement of the cavity diameter, ~1.6 nm, is determined in the correlation average (see Fig. 1c, *right*, inset).

conformational states. We also observed that the TMHs widened occasionally in the presence of ATP. Importantly, we characterized the working mechanism of Bcs1 by HS-AFM line scanning (HS-AFM-LS)[29,30] with microsecond temporal resolution. Our data shows that both ATP-binding and -hydrolysis coupled conformational changes are highly concerted, with an upper limit of the transition time coupling between neighboring protomers <30 microseconds, while the lifetime of the apo/ADP- and ATP-states are in the hundreds of milliseconds to the second timescale. Thus, Bcs1 is very different from other AAA ATPases with a concerted mechanism, likely an adaptation to its location in the IMM where a canonical AAA-ATPase action with hand-over-hand motions would lead to disruption of the IMM transmembrane potential.

## Results

### HS-AFM of Bcs1 in different nucleotide bound states

To visualize the dynamics of Bcs1 conformational changes by HS-AFM under close-to-physiological conditions, purified Bcs1 was reconstituted in membranes at low lipid-to-protein ratio. Then, the Bcs1 membranes were absorbed on freshly cleaved mica and imaged with HS-AFM in physiological buffer and at ambient temperature and pressure. The cryo-EM/X-ray structures indicated large structural differences between the IMS and the matrix sides (Fig. 1a, b), which provided us with the opportunity to assign the Bcs1 orientation in the HS-AFM images. Indeed, two different kinds of Bcs1 patches were found: Membranes exposing the large flower-shaped matrix face with high protrusion height, $<h_{matrix}> = 6.9 \pm 0.1$ nm (apo) (Fig. 1c), and membranes exposing small protrusions with low height, $<h_{IMS}> = 1.2 \pm 0.2$ nm (apo), separated by extended membrane areas (Fig. 1d, Supplementary Movie 1). All membranes that we found comprised uniformly inserted protein complexes, and we never found a mix of particles exposing the matrix and the IMS surface within one membrane. Likely the large soluble domains on the matrix surface provide a bias to the molecular orientation during the reconstitution process. Thus, based on comparisons of the structures and protrusion heights, we could unambiguously assign the two topographies to the matrix and the IMS surfaces, respectively. In apo and ADP conditions, the Bcs1 AAA-ATPase ring had a top-ring diameter of ~7.0 nm and an outer diameter of ~15 nm, i.e. corresponding to the center-to-center distances when the rings are densely packed (Fig. 1c, *left* and *middle*, Supplementary Fig. 1) in good agreement with the cryo-EM structure (Fig. 1b, *left* and *middle*). The seven subunits of the AAA-ATPase ring were clearly resolved in the raw data (Fig. 1c, *left* and *middle*) and enhanced in correlation averages (Fig. 1c, *insets*, *left* and *middle*). Thus, the substantial structural differences between the Bcs1 matrix and IMS faces allowed unambiguous assignment of sidedness in high-resolution HS-AFM images (Table 1).

The cryo-EM/X-ray data indicated that ATPγS triggered a substantial conformational change of the large C-terminal domain comprising the nucleotide binding domain (AAA-domain) (Fig. 1a, b, *right*)[24]. Thus, we next focused on the structure of the large C-terminal

domain in different nucleotide-bound conformational states (apo, ADP, ATPγS) by acquiring high-resolution HS-AFM movies. ADP-bound Bcs1 had an overall similar conformation as apo Bcs1 (Fig. 1c, *middle* and *left*, Supplementary Movie 2,3), which agreed well with the cryo-EM/X-ray data (Fig. 1a, b, *left* and *middle*)[23,24]. In the presence of ATPγS however, instead of a ring structure, we found a rather flat disk reaching a height of $6.2 \pm 0.1$ nm above membrane featuring seven peripheral protrusions (Fig. 1c, *right*, Supplementary Movie 4), which corresponded well to the small helical sub-domains of the AAA-domains that protrude into the matrix in the cryo-EM structure (Fig. 1a, b, *right*)[24]. Note, the central cavity in the ATPγS-bound state cryo-EM structure is only ~20 Å wide, which was barely visible in the HS-AFM raw data (Fig. 1c, *right*), but was resolved with a narrowed diameter in the correlation average (Fig. 1c, *inset*, *right*). It is important to note that the highest (brightest) parts of the apo and ADP-bound Bcs1 viewed by HS-AFM correspond to the RecA-like sub-domains of the AAA-domains (Fig. 1b, *left* and *middle*). Based on the cryo-EM/X-ray data, the diameter of the most protruding parts of the RecA-like domains is ~7.5 nm (Fig. 1b, *left* and *middle*)[24], which is consistent with our HS-AFM measurements, ~7.0 nm. The top-ring diameter of ATPγS-bound Bcs1 is ~10 nm in HS-AFM (Supplementary Fig. 1), corresponding to the distance between the AAA helical domains (Fig. 1b, *right*)[24]. Therefore, the diameter of the apo and ADP-bound Bcs1 appear smaller than that of ATPγS-bound Bcs1 in the HS-AFM topography, albeit the cryo-EM structures show that the ATPγS-bound state is overall slightly more compact (~13 nm) than the apo and ADP states (~14 nm). In brief, the overall topographic differences, different top-ring diameter (~10 nm *vs* ~7.0 nm), different protrusion height (6.2 nm *vs* 6.9 nm) and different appearance of the central cavity (~40 Å *vs* ~20 Å, often invisible in raw data), between ATPγS-bound and apo or ADP-bound states provided us the opportunity to unambiguously distinguish the conformational state of the Bcs1 complexes (Table 2), and next investigate their conformational dynamics during the ATPase cycle.

### Real time imaging of the Bcs1 conformational cycle in the presence of ATP

During HS-AFM movie acquisition of Bcs1 in the presence of ATP, dynamic structural changes of the Bcs1 rings were observed switching back and forth between two conformations of different heights and morphology (Fig. 2a, Supplementary Movies 5,6,7). In the low-height (dark) state, Bcs1 appeared relatively flat with a top-ring diameter of ~10 nm, which appeared well with the static structure of the ATPγS-bound conformational state observed by HS-AFM (Fig. 1c, *right*) and cryo-EM (Fig. 1a, b, *right*). Thus, we assigned the low-height state to the ATP-bound conformation. In contrast, the high-height (bright) state

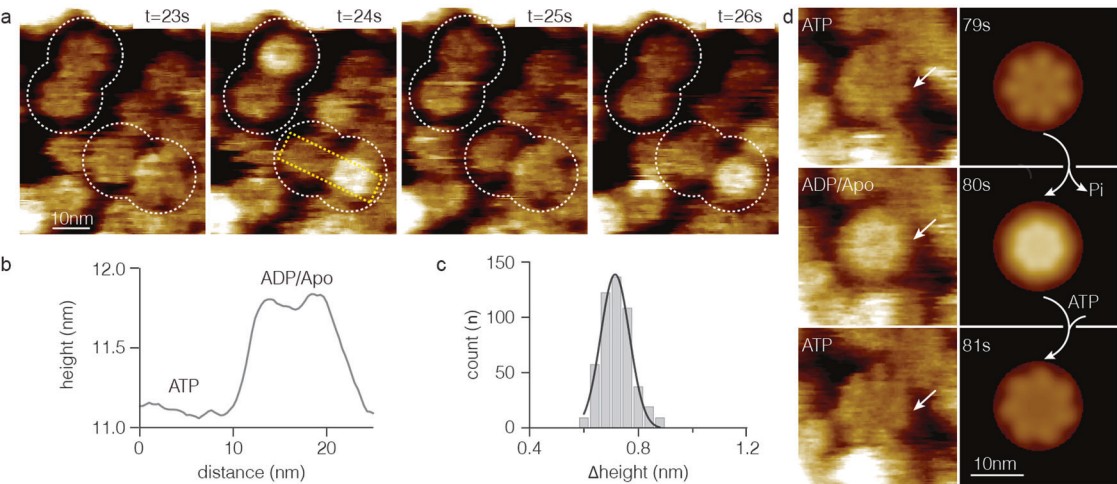

**Fig. 2 | HS-AFM time-lapse imaging of Bcs1 conformational changes in the presence of ATP. a** Dynamic structural changes of the Bcs1 matrix surface in presence of ATP. At $t = 23$ s, Bcs1 molecules (white dashed circles) are in ATP-bound conformation (low, dark). At $t = 24$ s, two Bcs1 complexes are in apo/ADP conformation (high, bright). At $t = 25$ s, the two Bcs1 that were in apo/ADP conformation switched back to the ATP conformation, and at $t = 26$ s, one of the Bcs1 rings reverted to the apo/ADP conformation (Supplementary Movie 5). **b** Height profile of Bcs1 rings along the yellow dash box showing the height difference between ATP and apo/ADP conformations, Δh -0.7 nm. **c** Height change distribution histogram of Bcs1 between ATP and apo/ADP conformations (line: Gaussian fit, $<\Delta h> = 0.72 \pm 0.08$ nm, $n = 497$). **d** HS-AFM images with subunit resolution (*left*, arrows) and 7-fold averaged images (*right*) of an individual Bcs1 during a conformational cycle. Arrows (*right*) indicate the enzymatic ATP-hydrolysis and -binding steps associated with the conformational changes. Source data are provided as a source data file.

was assigned to the apo and ADP-bound conformations (Fig. 1c, *left and middle*). As the apo and ADP-bound conformations are indistinguishable in our HS-AFM imaging, we term this state the apo/ADP conformation (though, given the affinity of ADP to the complex (see below), we are rather sure that this state is nucleotide-free and thus in the apo state in such turnover experiments) in these dynamic experiments where the Bcs1 complexes continuously bind and hydrolyze ATP. It is noteworthy that the height difference between ATP and apo/ADP conformations was -0.72 nm (Fig. 2b, c), in excellent agreement with the cryo-EM/X-ray data and the static HS-AFM imaging data of the apo, ADP, and ATPγS conformations (Fig. 1a, c, Table 2). Thus, even if the imaging resolution was not high enough to depict individual subunits, the substantial height difference between states allowed us to assign states to any position on the ring unambiguously. Importantly, high-resolution HS-AFM imaging allowed us to capture single Bcs1 complexes interchanging between ATP and apo/ADP conformations resolving individual subunits (Fig. 2d, Supplementary Fig. 2). Thus, we explored the working mechanism of Bcs1 in detail and at the single molecule level: All HS-AFM imaging data at 1 frame per second image acquisition were consistent with a mechanism where the seven subunits displayed concerted motions for both ATP-binding and -hydrolysis coupled conformational changes (Fig. 2d).

## HS-AFM line scanning provides Microsecond insights into the Bcs1 conformational cycle

Generally, the canonical, soluble, hexameric AAA-ATPases utilize a hand-over-hand working mechanism[18,20,21] where the protomers undergo sequential ATPase activity and conformational changes around a staircase-like ring, i.e., there is a transition time delay between neighboring protomers and at any given time many different nucleotide-binding and conformational states are found within the ring. In contrast to the hand-over-hand mechanism, another possible working mechanism would constitute a highly concerted conformational change, i.e., there is no transition time delay between neighboring protomers and all protomers undergo conformational changes at the same time. Because canonical AAA-ATPases work hand-over-hand, we reasoned that the HS-AFM imaging rate might just be too slow to see sequential protomer motion[31,32]. We thought that

increasing the temporal resolution would allow us to assign by which working mechanism Bcs1 worked. We further reasoned that, theoretically, to distinguish between sequential or concerted working mechanisms we only had to monitor the conformational changes of two protomers in the ring complex but at the fastest possible temporal resolution. Thus, we designed experiments more meticulously by introducing HS-AFM line scanning (HS-AFM-LS)[29,30]. In HS-AFM-LS, the slow scan axis (y-direction) is disabled (after targeting a single Bcs1 ring into the center of the observation area, Fig. 3a–d, top), and the central x-scan line is repeatedly scanned at a rate of up to a thousand lines per second. Then, the scan lines are stacked to generate kymographs (Fig. 3a–d, bottom), recording the dynamic behavior of biomolecules with millisecond temporal resolution, without losing z-dimensional spatial resolution which is important to assign conformational states. In our experiments, as Bcs1 is a heptameric ring complex and the AFM tip contours the two farthest positioned protomers in the ring when HS-AFM-LS is performed across the central pore (Fig. 3a–d, top, dashed lines), the kymographs are expected to have train-track appearance (Fig. 3a–d, *bottom*).

Apo and ADP-bound Bcs1 molecules appeared as ring structures in the HS-AFM imaging topography (Fig. 1c, *left* and *middle*, Fig. 3a, b, top), therefore, their corresponding kymographs appeared as two parallel bright lines contouring the two most lateral positioned protomers (Fig. 3a, b, *bottom right*, red and gray arrows). Tracking the topographic height along the two parallel lines (opposite protomers) the kymographs were transformed into height/time traces (Fig. 3a, b, *bottom left*). These protomer heights were consistently $<h_{apo}> = 6.9 \pm 0.1$ nm (apo) and $<h_{ADP}> = 6.9 \pm 0.1$ nm (ADP). In the case of ATPγS-bound Bcs1 that appeared disk-shaped in imaging (Fig. 1c, *right*, Fig. 3c, top) the kymograph appeared bar-like without a central indentation (Fig. 3c, *bottom right*). Translating the kymograph peripheral topography into height/time traces resulted in $<h_{ATP\gamma S}> = 6.2 \pm 0.1$ nm (ATPγS) (Fig. 3c, *bottom left*). Thus, the ATPγS conformation was unambiguously distinguishable from the apo and ADP-bound Bcs1 also in kymographs.

Investigating Bcs1 in the presence of ATP (Fig. 3d, top), the HS-AFM-LS kymographs displayed back-and-forth switching between ATP-bound (low) and apo/ADP-bound (high) conformational states (Fig. 3d,

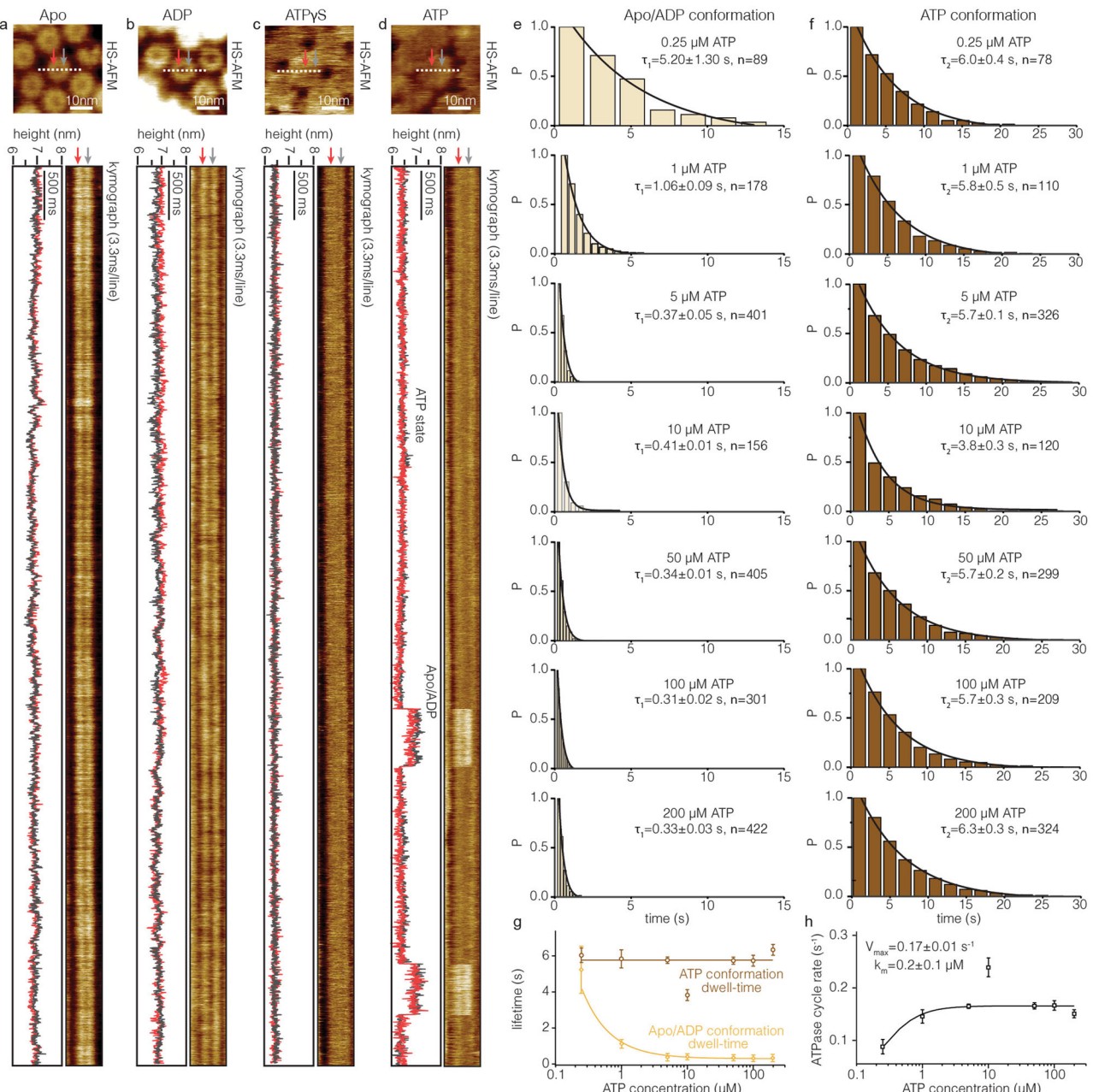

**Fig. 3 | HS-AFM-LS of Bcs1 at millisecond temporal resolution. a** Top: HS-AFM image of Bcs1 in apo conformation. White dashed line represents HS-AFM-LS kymograph position. Bottom right: HS-AFM-LS raw data kymograph at 3.3 ms temporal resolution. Gray and red arrows represent the topography of the two most lateral positioned protomers. Bottom left: Height/time traces of the protomers in the kymograph. **b**–**d** HS-AFM-LS experiments in the presence of ADP **b**, ATPγS **c**, and ATP **d**, respectively. **e**, **f** Dwell time distributions (see Methods) of the apo/ADP- **e** and ATP- **f** conformations at various ATP-concentrations. The dwell time distributions were well-fitted by a single exponential. **g** Dwell times of the apo/ADP (*yellow*) and ATP (*brown*) conformations as a function of ATP-concentrations.

The error bar at each ATP concentration was determined through exponential fitting of the corresponding dwell-time histograms in **e**, **f**. (Apo/ADP conformation dwell time: $n = 89$ for 0.25 μM, $n = 178$ for 1 μM, $n = 401$ for 5 μM, $n = 156$ for 10 μM, $n = 405$ for 50 μM, $n = 301$ for 100 μM and $n = 422$ for 200 μM. ATP conformation dwell time: $n = 78$ for 0.25 μM, $n = 110$ for 1 μM, $n = 326$ for 5 μM, $n = 120$ for 10 μM, $n = 239$ for 50 μM, $n = 209$ for 100 μM and $n = 324$ for 200 μM). **h** ATP concentration dependence of the Bcs1 conformational turnover rate. The turnover rate, along with its associated error bar, was extracted from the data presented in **g**. Similar results as in **a**–**d** were observed in at least 5 independent biological experiments. Source data are provided as a source data file.

*bottom right*), which is consistent with our high-resolution movies (see Fig. 2, Supplementary Movies 5,6,7) but with ~300 times higher temporal resolution. The height difference between conformational states in the height/time traces was $<\Delta h> = 0.7 \pm 0.1$ nm (Fig. 3d, *bottom left*, Fig. 4a, *right*), in excellent agreement with the cryo-EM and HS-AFM imaging data (Fig. 2c, Table 2). From these height/time traces, using state assignment algorithms, the characteristic lifetimes of the apo/ADP (Fig. 3e) and the ATP (Fig. 3f) conformational states could

be determined at various ATP-concentrations. The dwell time for the ATP conformational state, $\tau_{(ATP)}$, is ~5.6 s and independent of the ATP-concentrations (Fig. 3g), while the dwell time for the apo/ADP conformational state, $\tau_{(apo/ADP)}$ shortened with increasing ATP-concentration, tending to plateau at 0.32 s at 5 μM or higher ATP-concentration (Fig. 3g). The maximum turnover rate, $V_{max}$, is ~0.17 s$^{-1}$ and the Michaelis constant, $k_m$, is ~0.2 μM (Fig. 3h). This corresponds to an ATP turnover of 1.2 ATP per second per Bcs1 ring. This is in excellent

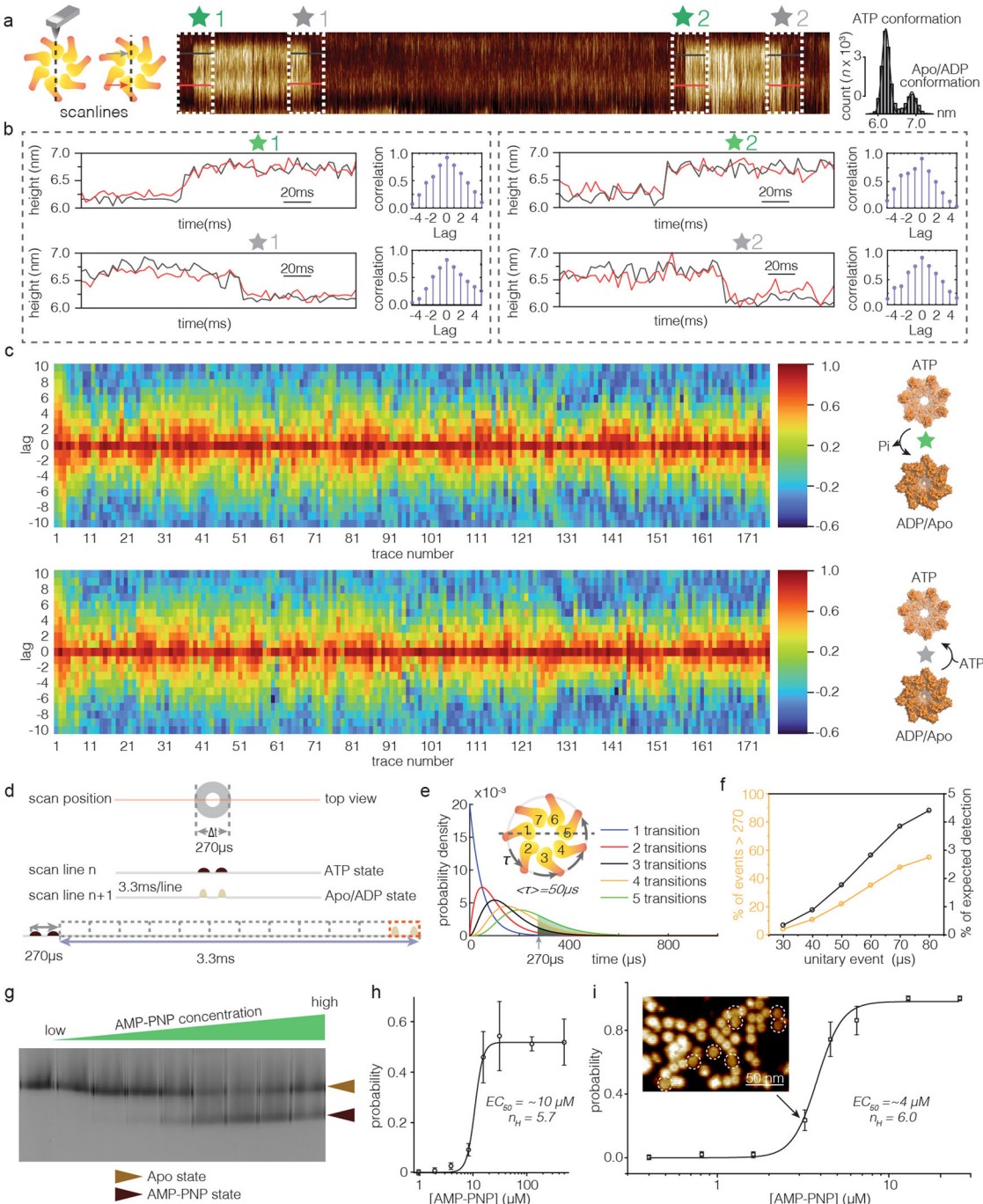

**Fig. 4 | Conformational changes of Bcs1 at microsecond temporal resolution.**
**a** Left: HS-AFM-LS schematic. Middle: HS-AFM-LS kymograph of Bcs1 in the pre-
sence of ATP. Green and gray stars are the time point where ATP-hydrolysis and
ATP-binding related conformational changes occur, respectively. Right: Height
distribution analysis of the kymograph in ATP-conditions. The Δheight between the
apo/ADP and ATP conformations is -0.7 nm. **b** Time-lag cross-correlation analysis
of ATP to apo/ADP (green star) and apo/ADP to ATP (gray star) conformational
changes, respectively. The correlation peak indicates the time-lag when the two
traces are most synchronized (1 lag = 3.3 ms, the HS-AFM-LS temporal resolution).
**c** Correlation plots for all ($n = 178$) traces, ATP to apo/ADP (*top*), apo/ADP to ATP
(*bottom*) conformational changes. The correlation peak is at 0 time-lag for all tra-
ces. **d** Schematic illustration of HS-AFM-LS across Bcs1 during ~270 μs. Below: Both
Bcs1 peripheral protomers are in ATP (scan line n) and apo/ADP (scan line n + 1)
conformation. Bottom: Dividing the 3.3 ms scan line time into 270 μs windows
(dashed boxes) indicated that the Bcs1 conformational change should coincide in

8% of the events with the tip crossing the complex (red dashed box). **e** Convolution
of probability distributions with unitary τ = 50 μs for 1 (blue line), 2 (red line), 3
(black line), 4 (orange line), and 5 (green line) transitions. The black, orange and
green shaded areas represent events take longer than 270 μs for 3, 4 and 5 transi-
tions, respectively. **f** Probability of occurrence of events >270 μs for 4 transitions
with unitary event values ranging from 30-80 μs. Right axis: Corresponding theo-
retical detection probability in our HS-AFM-LS traces. **g** Blue-Native page gel of Bcs1
as a function of AMP-PNP concentration. Top bands represent apo- and bottom
bands AMP-PNP conformation Bcs1. **h, i** Probability of AMP-PNP conformation as a
function of the AMP-PNP concentration derived from BN-PAGE gel **h** and HS-AFM
**i** experiments, respectively. The error bars were derived from 3 biologically inde-
pendent experiments. (lines: best fit of Eq. (2), white dash circles in the HS-AFM
image **i** represent the molecules in AMP-PNP conformational state. EC50: half
maximal effective concentration, $n_H$: Hill coefficient). Source data are provided as a
source data file.

agreement with bulk ATPase activity experiments that reported 0.7 ATP per second per Bcs1 ring[24], and highlights that the complex can operate at full speed in our experimental setting.

## Concerted conformational changes in the Bcs1 ring

To analyze the speed of the conformational changes in detail, i.e., to decipher whether Bcs1 operated by the canonical hand-over-hand mechanism or by a concerted mechanism, we focused on the height change steps in the kymographs for both ATP-hydrolysis (Fig. 4a, *green asterisks*) and ATP-binding (Fig. 4a, *gray asterisks*) coupled conformational changes. Benefitting from the high spatial resolution of HS-AFM-LS, the two Bcs1 protomers located opposite in the ring complex appeared as two parallel lines when contoured by the HS-AFM tip, which were individually converted into height/time profiles with high signal-to-noise ratio (Fig. 4b). Thus, we next analyzed the synchronicity of the protomer conformational changes by analyzing time-lag cross-correlation (TLCC) between the two height/time profiles[33]. This analysis resulted in correlation plots as a function of time-lags (1 lag = 3.3 ms, the HS-AFM-LS rate, Fig. 4b and Supplementary Fig. 3). The correlation peak indicates the time-lag at which the two traces are most synchronized, i.e., zero-lag of the correlation peak indicates that the two protomers undergo the conformational change at the same time, in a concerted way (Fig. 4b, *right*). Intriguingly, we found that the correlation peak was always at time-lag *offset* = 0, in all 178 analyzed traces (Fig. 4c), which indicated that the two protomers located on opposite sides of the Bcs1 rings underwent their conformational change simultaneously for both ATP-hydrolysis (Fig. 4c, top) and ATP-binding (Fig. 4c, *bottom*) at a temporal resolution of 3.3 ms.

Next, we reasoned that the zero-lag-time detection in the HS-AFM-LS traces could allow us to narrow the conformational concertation between Bcs1 protomers even further down to the microsecond level, because we know the speed at which the tip crosses the Bcs1 rings and probes the two peripheral protomers. Taking as example the ATP-hydrolysis induced conformational change (Fig. 4d): If in a HS-AFM-LS kymograph, both Bcs1 protomers on opposite sides of the ring were in the ATP conformation in scan line (n), and both were in the apo/ADP conformation in scan line (n + 1), then the conformational change must have happened in both protomers at any time between the two scan lines (within the 3.3 ms). However, given the size of the Bcs1 complex and the speed of the HS-AFM tip, we know that the time duration scanning across the Bcs1 ring was 270 µs (see Methods). Thus, the probability of the conformational change to happen in the time window when the tip crossed the Bcs1 ring is ~8% (270 µs/3300 µs) (Fig. 4d, *red dashed box*). Thus, if the conformational transition time between the two protomers was longer than 270 µs, we should detect ~8% (~ 15 events among the 178 analyzed traces) of non-synchronized height/time traces in our HS-AFM-LS experiments—but we never did. To ascertain our rationale, we performed numerical simulations, in which the transition time between detecting two points and the scan time resolution were set to 270 µs and 3.3 ms, respectively, and confirmed the calculated detection probability (Supplementary Fig. 4). Thus, we can set the upper limit of the conformational coupling to 270 µs. We think that this is strong evidence for a concerted power stroke mechanism in Bcs1, as conformational changes of smaller amplitude for example in transmembrane transporters operate in the millisecond range[29,34]. We did however not rule out a sequential model for ATP-hydrolysis and -binding coupled conformational changes. Considering that the transition within the ring was a sequence of unitary events with a given time constant, ~4 such transition events had to occur from the protomer on one side to the protomer on the other side of the Bcs1 ring (Fig. 4e, *inset*). Thus, a convoluted time probability distribution of *n* transitions could be calculated from the unitary event with a given time constant described by a single exponential[35] (Fig. 4e, blue line, see Methods). From experiment we know that the probability for these ~4 sequential transitions to occur should be shorter than 270 µs,

allowing us to estimate the unitary transition time constant (Fig. 4e and Supplementary Fig. 5). Using unitary transition time constants of 30 µs, 40 µs, 50 µs, 60 µs, 70 µs or 80 µs, we found that the probability for 4 transitions to occur over >270 µs was ~4%, 11%, 22%, 35%, 48% or 55%, respectively (Fig. 4f, *orange line*). Considering the experimental zero-lag-time detection probability of ~8% to detect an event longer than 270 µs, transition composed of unitary time constants of 30 µs or 40 µs should be detected ~1 or ~2 times among our 178 analyzed traces (Fig. 4f, *black line*). Thus, if the ATP-hydrolysis and ATP-binding coupled conformational transitions occurred sequentially, the neighbor transition coupling time should be faster than 30 or 40 microseconds.

From our HS-AFM-LS experiments (Fig. 4a) we know that Bcs1 stays in the apo/ADP conformation for quite a long time under saturating ATP concentrations, $<\tau_{(apo/ADP)}>$ ~0.32 s (Fig. 3g), and then the complex undergoes a concerted apo/ADP- to ATP-conformational change within <270 µs (Fig. 4c, *bottom*). Thus, the duration of the complex being in the apo/ADP state is >1185 times longer than the ATP-binding related conformational change in the complex. How can this be explained, and how do these data relate to ATP-binding to the complex? To get further insights how ATP-binding relates to the conformational change, we took advantage of a particularity of the Bcs1 complex in blue native polyacrylamide gel electrophoresis (BN-PAGE) gel migration. Indeed, the difference between the apo and ATP-bound (AMP-PNP on the gel) conformations are so substantial that the complex migrates differently in BN-PAGE gels[24]. Accordingly, we ran the complex in BN-PAGE at increasing AMP-PNP concentrations and analyzed the intensities of the apo- and AMP-PNP-complex bands (Fig. 4g, Supplementary Fig. 6). Plotting the probability of occurrence of Bcs1 in AMP-PNP-conformation as a function of AMP-PNP concentration we found an $EC_{50}$ of ~10 µM and derived a Hill coefficient, $n_H$ = 5.7, thus close to the seven subunits in the complex (Fig. 4h)[36]. We note that even at very high AMP-PNP concentration we did not reach a 100% conformational conversion in the BN-PAGE. We think this is either due to traces of ADP in the solution, or single protomer damages in the Bcs1 rings in the BN-PAGE. Both explanations favor a highly concerted conformational change as either occupation of a subunit with ADP or damage of a single subunit would prohibit the entire ring from transitioning into the AMP-PNP-conformation. We also performed HS-AFM imaging-based experiments to assess the probability of Bcs1 to be in the AMP-PNP conformation as a function of AMP-PNP concentration (Fig. 4i). Encouragingly, this analysis revealed an $EC_{50}$ of ~4 µM and a Hill coefficient $n_H$ = 6.0, which closely matched the values determined through BN-PAGE gel analysis. In brief, the experimental data, the mathematical model extending the analysis of the experimental data, the apo/ADP state dwell-time analysis combined with the conformational transition analysis, and the BN-PAGE analysis, all point to highly concerted ATP-binding and conformational changes.

## Conformational changes on the Bcs1 IMS face

Having acquired a clear understanding of the conformational dynamics of the large C-terminal domain facing the matrix side, we next focused on the dynamics of the Bcs1 IMS side in presence of ATP. In apo conditions the IMS side appeared as a small dome-shaped protrusion with a height of ~1.2 nm (see Fig. 1e). Interestingly, we found that in the presence of ATP the IMS side, alike the matrix side, switched dynamically between two conformations with different heights and widths (Fig. 5a, Supplementary Movie 8), one matching the apo observations and one with increased height and diameter. As the increases in width and height correlated (Fig. 5a, *bottom*), we analyzed the conformational changes of the IMS side as a function of time by measuring the protrusion volume, which had a higher signal than either height or width alone. In presence of ATP, we found spikes in the volume/time traces with values of >20 nm³ (Fig. 5b), which were not detected in apo conditions (Fig. 5c). Accordingly, the volume distribution in presence of ATP had a long tail to larger volumes, which

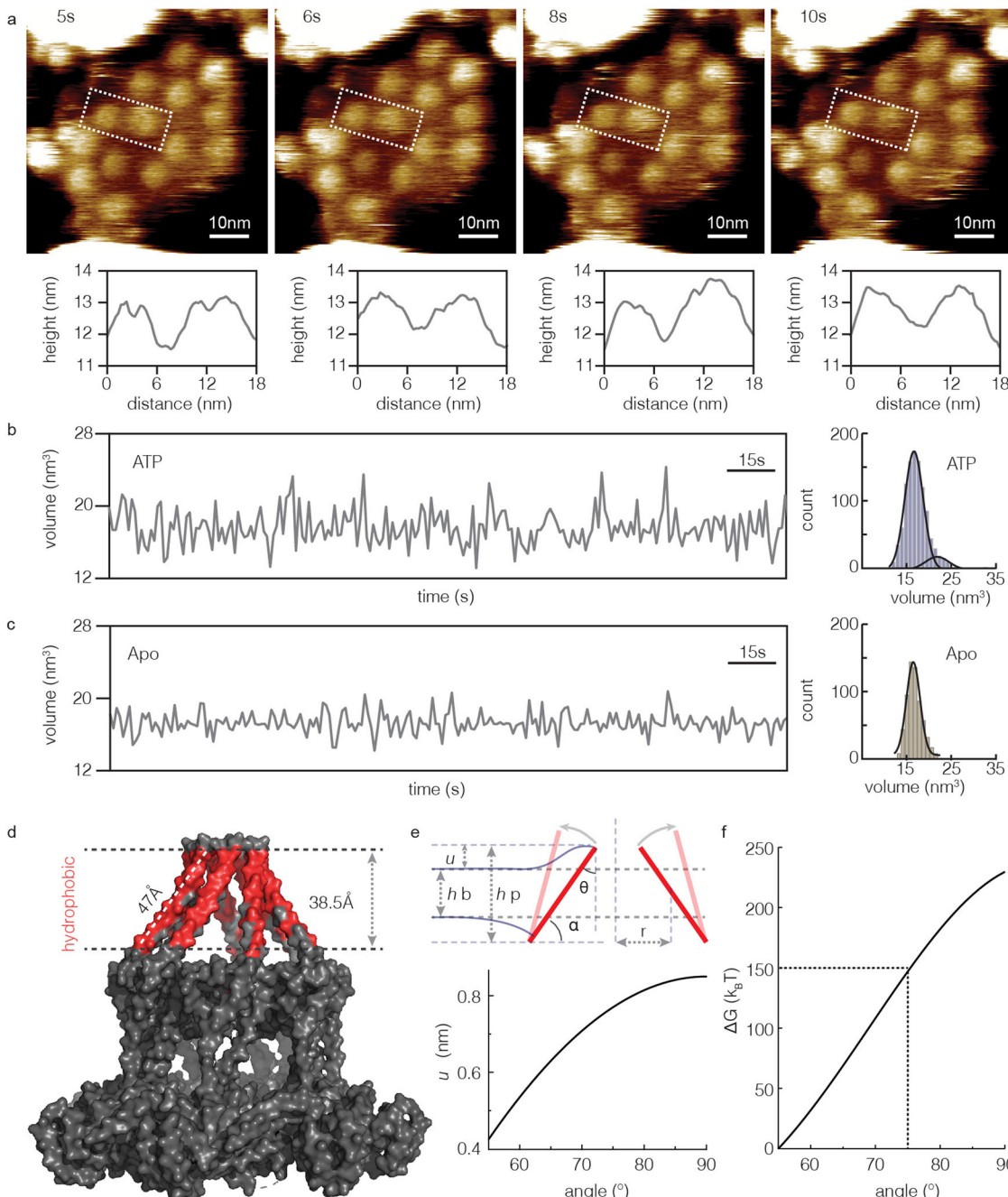

**Fig. 5 | Conformational changes of the Bcs1 IMS face opening the protein transport gate. a** Top: HS-AFM movie frames showing conformational changes of the Bcs1 IMS face in the presence of ATP. Bottom: Height profiles along the white dashed boxes showing the topography of two neighboring complexes. **b, c** Left: Single molecule Bcs1 IMS face volume/time traces in ATP- **b** and in apo- **c** conditions. Right: Volume distributions in ATP- **b** and in apo- **c** conditions. Lines: Gaussian fits. The volume distribution in ATP-condition is best fitted by two Gaussians. **d** Analysis of the TMHs hydrophobic thickness (PDB 6SH4). **e** Top:

Schematic of bilayer deformations in a scenario where the TMHs straighten in the membrane. µ represents the hydrophobic mismatch, where $hb$ and $hp$ represent the hydrophobic thickness of Bcs1 TMH and lipid bilayer, respectively. α is the angle between TMH and an axis normal to the membrane plane. θ is the membrane slope angle at the helix, $\theta = 90°\text{-}\alpha$. r is the protein gate radius. Bottom: Hydrophobic mismatch change as function of α. **f** Change in free energy profile as a function of α (Eq. (6)). Source data are provided as a source data file.

was well fitted by a bimodal Gaussian function, where the first fit peaked at ~16 nm³ ($v_1$), while the second peaked at ~21 nm³ ($v_2$) (Fig. 5b, *right*). In apo conditions the volume distribution was well fitted by a single Gaussian peaking at ~16 nm³ ($v_1$) (Fig. 5c, *right*). Quantitative analysis of the $IMS_{(ATP)}$ volume distribution showed that only ~5% of the molecules were in the $v_2$-population with larger volume representative of the open conformation (Fig. 5b, *right*). Given that the $v_2$-spikes on the IMS face are so rare, ~5%, and short lived (<1 s, Fig. 5b,

*left*), while the matrix face is ~90% of the cycle in the ATP-bound conformation and for extended time, $<\tau_{(ATP)}> = ~5.6$ s (see Fig. 3f,g), we suggest that the open conformation of the IMS side is related to ATP-hydrolysis or Pi release on the matrix side and represents a short-lived transition at the end of the ATP-bound conformational state.

To allow the ISP transport (its globular C-terminal domain of 126 residues has an estimated diameter ~3.5 nm), the pore seal radius of the TMHs should increase to ~1.7 nm. Given that the TMHs are ~5 nm long

and form a closed cone with a tilt angle $\alpha$ of 55° in the apo state (Fig. 5d), we calculate that $\alpha$ should change to ~75°, straightening the helices in the membrane, to result in a pore radius of ~1.7 nm (5 nm (cos55°–cos75°)) (Supplementary Fig. 7). Alternatively, a lateral displacement where the TMHs open a pore according to an iris diaphragm-like mechanism is also conceivable, though in this case the increase in height detected by HS-AFM would not be accounted for. In the first case, when the TMHs straighten up in the membrane to open the pore, not only the membrane is pushed aside to allow the formation of an open gate, but also the hydrophobic thickness of the TMH region would change from 38.5 Å to 45.3 Å, resulting in a hydrophobic mismatch between protein and membrane, and thus membrane deformation (Fig. 5d,e) and an associated free energy change (Eq. 5f, Eq. (6))[37–39]. Using membrane elastic theory, we estimated the deformation free energy change as function of the TMH opening angle $\alpha$ from 55° to ~75° of ~150 $k_{B}T$ (Fig. 5f, Methods). In a scenario where the helices moved iris-like, only the in-plane area expansion of the protein against the membrane would have to be considered, giving an estimate of ~15 $k_{B}T$ (Eq. (6)). Obviously, any scenario in between these two extremes of helical motions is conceivable and likely. Notwithstanding these estimates, such an energy cost estimation is meaningful for the elucidation of the functional role of the TMH cone cavity as well as regarding a concerted ATP-hydrolysis coupled conformational change (see Discussion).

## Discussion

In this work, we applied HS-AFM and HS-AFM-LS to characterize Bcs1 in different nucleotide conditions (apo, ADP, ATP, ATPγS) under close-to-physiological conditions. We found that ADP-bound Bcs1 shared overall similar structure as in apo conditions, while ATPγS triggered an obvious conformational change (Fig. 1). In the presence of ATP, Bcs1 ATPase conformational cycles were observed in both HS-AFM imaging and HS-AFM-LS (Figs. 2 and 3). We analyzed the protomers along HS-AFM-LS scan lines at two opposite sides of the Bcs1 ring and found that their actions were synchronized (Fig. 4). In saturating ATP-conditions Bcs1 stayed in the apo/ADP state for ~0.32 s, after which the protomers underwent a concerted structural change within 270 μs or shorter. Following, Bcs1 remained in the ATP-conformational state for quite long, ~5.6 s, after which the protomers also underwent a concerted conformational change within 270 μs or shorter to revert to the apo/ADP conformation. While HS-AFM-LS allowed us to observe the conformational changes at high temporal resolution and provided strong evidence for a concerted conformational change within the Bcs1 ring, HS-AFM cannot see the nucleotide-binding state inside the protomers. How are the apo, ATP, ADP + $P_i$, ADP nucleotide-states related to the conformational states: Is ATP-binding and -hydrolysis also concerted and matching the conformational changes?

We may consider different scenarios for the ATP-binding step (Supplementary Fig. 8a): (i) The conformational change from the apo/ADP- to the ATP-state is directly coupled to ATP-binding. In this scenario, all 7 subunits are empty at the end of the 0.32 s apo/ADP-dwell, and then bind all 7 ATPs within <270 μs leading to the conformational change. (ii) During the 0.32 s apo/ADP-dwell each subunits binds an ATP one-by-one, but only when the 7th subunit binds ATP the entire ring transfers into the ATP conformation. (iii) During the 0.32 s apo/ADP-dwell the ATPs bind and unbind sporadically to the ring, and stochastically the apo/ADP- to ATP-conformational change occurs. Then, in the ATP-conformation the affinity for ATP is much higher, and all subunits swiftly bind ATP. Our BN-PAGE and HS-AFM analyses indicate a high Hill coefficient, $n_H$ = ~6, to reach the AMP-PNP conformation, in possible agreement with all three scenarios. However, from a structural perspective, analyzing the nucleotide binding sites in the apo, ADP and ATPγS conformations, we note that the nucleotide-binding site in the ATPγS conformation is substantially different from the other conformations (RMSDs: ATPγS vs apo: 10.1 Å, ATPγS vs ADP:

10.5 Å, ADP vs apo: 1.8 Å, Supplementary Fig. 9) indicative that the ATPγS conformation has an increased affinity for ATP, and therefore we propose that scenario (iii) is the most likely. Further work is needed to test this hypothesis.

We also consider different scenarios for the ATP-hydrolysis step (Supplementary Fig. 8b): (i) All 7 ATP molecules are hydrolyzed and release Pi in a highly concerted way in the last <270 μs of the ATP-conformation dwell, followed by the conformational change to the apo/ADP conformation. (ii) During the 5.6 s ATP-conformation dwell the 7 ATP molecules are hydrolyzed one-by-one, but only hydrolysis of the 7th ATP permits the conformational transition. (iii) The 1st subunit that hydrolyses ATP induces the conformational transition that leads to a highly concerted ATP-hydrolysis and Pi-release in all subunits. Our HS-AFM-LS analysis shows that the ATP conformation dwell is ATP concentration independent, which is in disagreement with scenario (ii) where ATP from the bulk would continuously replace ADP in the ATP conformation binding pockets (Supplementary Fig. 10,11). Scenarios (i) and (iii) are very similar, both implying a tight coupling between ATP-hydrolysis and the ATP- to apo/ADP conformational transition. Again, we favor scenario (iii) based on the differences of the binding pockets between conformations (Supplementary Fig. 9). Membrane elastic theory estimates that an energy cost of order of tens of $k_{B}T$ is needed to tilt the TMHs, push the lipids away and open the gate (a ~ 3.5 nm diameter pore in the membrane is needed for ISP release into the IMS), which would need a concerted hydrolysis-coupled action.

Generally, the individual protomers of AAA-ATPases, such as Vps4[20] or Cdc48[21,40], interchange between the three different nucleotide states (ATP, ADP, apo) in a sequential manner and where the three different nucleotide states and conformations always co-exist within a staircase-like ring complex. Based on these structures, the canonical hand-over-hand mechanism has been proposed. However, recent studies have proposed different models for ClpXP, despite their similar structures to the other AAA-ATPases[41–43]. Additionally, although the Abo1 AAA+ domain shares a structural fold resembling other canonical AAA-ATPases arranged in a staircase-like manner, HS-AFM data indicated that Abo1 subunits may stochastically hydrolyze ATP without an ordered sequence[28]. These studies show the limitations of determining the exact mechanism solely based on the structure. Here, by analyzing the Bcs1 ATPase cycle using HS-AFM imaging and HS-AFM-LS with much increased temporal resolution, we revealed a so far unseen highly concerted conformational mechanism where the subunits in the ring have the same conformation over extended dwell times and then undergo concerted conformational changes in a burst-like manner. We propose two main physiological reasons why this is important: First, to translocate a folded globular ISP (diameter ~3.5 nm) across a seal formed by the conically arranged TMHs a free energy change of order ~150 $k_{B}T$ is expected. Given that the hydrolysis of 1 ATP provides 15–30 $k_{B}T$, depending on the physiological conditions[44–46], the energy from the hydrolysis of all 7 ATP molecules in a concerted way (105–210 $k_{B}T$) would account for such an energy cost. Second, most of the time the Bcs1 complex must be kept tightly shut to avoid the devastation of the transmembrane potential. Indeed, its location in the inner mitochondrial membrane that maintains $H^+$, ion and solute gradients for energy generation and transport, is incompatible with the canonical processive AAA-ATPase action. According to our model, the long hydrophobic helices only open shortly through the concerted action of the entire ring complex and would otherwise be tightly sealed by the large energy cost associated with increased hydrophobic mismatch, membrane bending and occupied membrane area when adopting the open state.

Thus, a concerted ATP-hydrolysis coupled conformational change would provide sufficient energy to straighten or twist the TMHs in the membrane and open a pore wide enough for ISP translocation to the IMS side. Also, this model of action would provide solution for another

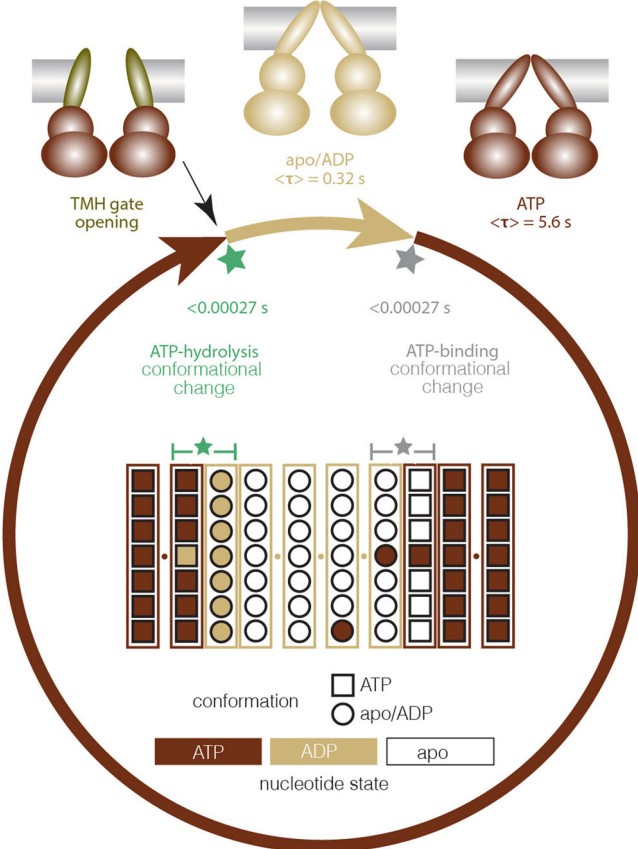

**Fig. 6 | Bcs1 working cycle.** Bcs1 stays in the apo/ADP-conformation for ~0.32 s. Following ATP-binding, it will transition to the ATP-conformation within <270 μs and remain in the ATP-conformation for ~5.6 s, until it reverts to the apo/ADP-conformation again within <270 μs. The IMS pore briefly opens at the end of the ATP-bound conformation. Center: Proposed nucleotide state before, during, and after the concerted conformational transitions (see Discussion, Supplementary Fig. 8).

problem: ISP has not only a globular head, but also a transmembrane helix which needs to be released from the Bcs1 translocation machinery. The straightening of the Bcs1 TMHs would not only open the gate at the IMS surface for the ISP globular domain to exit to the IMS, but also spread the Bcs1 TMHs for the ISP TMH to pass laterally into the membrane. Indeed, the Bcs1 TMHs are separated by ~3.5 nm at the bottom of the cone (matrix membrane surface) and their straightening would create gaps >1 nm large enough for a helix to laterally pass into the membrane. The N-terminal tail of ISP that faces the matrix has no secondary structure (see PDB 1SQB) and likely can sneak out between the open TMHs at the matrix membrane surface.

Overall, our results point towards the following mechanism (Fig. 6): Bcs1 interchanges between the two conformational states apo/ADP and ATP, that have extended dwell-times. Highly concerted actions to interchange between these conformations occur at the end of these dwells. Our data further indicates that in addition to the apo/ADP- to the ATP-bound conformations (both described by cryo-EM) a third so-far-elusive and short-lived conformation should exist with wide open TMHs. This conformation should briefly occur at the end of the ATP-conformation dwell to expel the cargo to the IMS. Our results show that Bcs1 has a different action mechanism as compared to other AAA-ATPases, with highly concerted conformational changes and extended dwells where all subunits in the ring complex have the same conformation and nucleotide state. This particularity seems like a logical adaptation to its function as a folded protein transporter and its location in the inner mitochondrial membrane.

## Methods

### Protein expression and purification

The cDNA clone of mBcs1 (accession number BC019781) was purchased from Life Technologies. The mBcs1 code region (residues 1–418) was cloned into the Pichia yeast expression vector pPICZ A (Life Technologies). The hexa-histidine tag was put in the C-terminal of full-length mBcs1 (pPICZ-mBcs1-His) for purification. Then, the pPICZ-mBcs1-His plasmid was linearized by restriction enzyme PmeI before being introduced to Pichia by electroporation. For mBcs1 expression, *Pichia pastoris* yeast transformed with linearized pPICZ-mBcs1-His plasmid was grown in minimal glycerol medium (1.34% yeast nitrogen base (YNB), 1% glycerol, $4 \times 10-5\%$ biotin) at 29 °C. At OD600 = ~4, cells were transferred in minimal methanol medium (1.34% YNB, 4 × 10−5% biotin, 0.25% methanol) with OD = 1, and were cultured at 29 °C for 4 d, with supplement of 0.25% methanol every 24 h.

Cells were collected by centrifugation (4,000 g, 25 min) and re-suspended in homogenization buffer (100 mM Tris, pH 8, 100 mM sucrose, 1 mM EDTA, 2 mM phenylmethylsulfonyl fluoride) with a final concentration of ~0.3 g/ml and disrupted by a Microfluidizer (Microfluidics International Corporation) at 2500 bar. Cell debris was spun down at 3,500 g for 25 min. Then, the supernatant was centrifuged at 125,000 g for 1 h (4 °C) to collect membrane pellets. The membrane pellets were washed in homogenization buffer to a concentration of 0.5 g/ml and pelleted again by centrifugation (125,000 g, 4 °C, 1 h) three times. Finally, the crude membranes were re-suspended in a solubilization buffer (25 mM potassium phosphate, pH 7.4, 200 mM NaCl). Protein concentration was determined using the bicinchoninic acid assay method[47] (Thermo Fisher Scientific).

The crude membranes with a protein concentration of 5 mg/ml were incubated with 0.5% CHAPS for 30 min at 4 °C. Insoluble components were removed by centrifugation (125,000 g, 4 °C, 30 min). The solubilized mBcs1 was mixed with Ni-NTA resin pre-equilibrated with buffer A: 25 mM Tris, pH 8, 300 mM NaCl, 10% glycerol, 10 mM imidazole, 0.05% N-dodecyl-β-d-maltoside (DDM) (Anatrace) for 30 min at 4 °C. Then, the Ni-NTA resin was loaded into a column and washed with buffer A supplemented with 100 mM imidazole. mBcs1 was eluted with buffer A containing 250 mM imidazole. mBcs1 was concentrated using a 100 kDa cut-off Amicon ultra concentrator (MilliporeSigma) and further purified by gel filtration (Superdex 200, GE Life Sciences, in 20 mM Tris, pH 8.0, 200 mM NaCl, 10% glycerol, 0.05% DDM).

### Protein reconstitution

Purified Bcs1 was mixed with soy extract polar lipid (Avanti) at low lipid to protein ratio (LPR) of ~0.5 (wt/wt) and complemented with a buffer containing 10 mM HEPES (pH 7.5), 200 mM NaCl, 2.5 mM NaN3, and 0.02% DDM to a final protein concentration of 1 mg/ml. The mixture was equilibrated for 4 h, followed by adding biobeads for detergent removal overnight at 4 °C.

### HS-AFM

Two microliters of reconstituted Bcs1 membranes were deposited on a 2 mm diameter freshly cleaved mica for ~10 min, followed by gentle rinsing with imaging buffer (10 mM HEPES, pH 7.5, 200 mM NaCl and 2 mM MgCl₂). The different nucleotide bound states (ADP, ATPγS, AMP-PNP, and ATP) were imaged in imaging buffer containing 1 mM ADP, 10 μM ATPγS, AMP-PNP ranging from 0.4 μM to 25 μM and various ATP concentration ranging from 0.25 μM to 200 μM, respectively. All movies were acquired using a HS-AFM (SS-NEX; RIBM) microscope operated in amplitude modulation mode, using a lab-built amplitude detector[48] and a free amplitude and force stabilizer[49]. Ultra-short cantilevers (NanoWorld) with a nominal spring constant of 0.15 N/m and resonance frequency of ~0.7 MHz in liquid were used. All the Bcs1 high resolution movies were acquired at a scan rate of 1 frame per second and 300 × 300 pixels.

For HS-AFM line scanning (HS-AFM-LS), Bcs1 molecule was centered in the image scan before switching to HS-AFM-LS mode. HS-AFM-LS kymographs were collected at a rate of 3.3 ms per left-to-right and right-to-left scan motion. Given that the kymographs are composed of left-right scan lines only, each line was acquired during 1.67 ms, but each identical position is probed every 3.3 ms. Considering the size of the Bcs1 molecule is ~8 nm and the line scan size of 50 nm, the duration of scanning over one Bcs1 ring is ~270 μs.

## Microscale thermophoresis (MST) experiments

MST experiments were performed using a Monolith NT.115 Pico (NanoTemper Technologies, Germany) and Monolith NT.115 Premium capillaries (NanoTemper Technologies, Germany). The Bcs1 complex was labeled using the Monolith Protein Labeling Kit RED-tris-NTA, 2nd Generation. For the measurements, the concentration of labeled Bcs1 was kept constant (50 nM), and the concentration of ADP was varied from 0.007862 μM to 62.5 μM, and the concentration of phosphate ions was varied from 0.00122 mM to 40 mM. All experiments were performed in measuring buffer (10 mM HEPES, pH 7.5, 200 mM NaCl, and 2 mM MgCl$_2$, 0.05% DDM). The MST traces were analyzed using the software M.O. Affinity Analysis v2.3 (NanoTemper Technologies, Germany).

## Data analysis

All the movies were aligned in ImageJ using laboratory-build plugins. The center-to-center distance of neighbor particles was analyzed in ImageJ with laboratory-build programs. The top-ring diameter, cavity diameter, protrusion height and width of IMS side were analyzed in ImageJ. Knowing protrusion height ($h$) and width ($w$), the volume ($v$) could be estimated by

$$v = \frac{2}{3}\pi\left(\frac{w}{2}\right)^2 h \qquad (1)$$

To analyze the dwell-time for each state, the kymographs were processed using the Step Transition and State Identification (STaSi) routine programmed in MATLAB[29,50]. To analyze the synchronicity of the two peripheral protomers, the kymographs were analyzed by time-lagged cross-correlation (TLCC), which has been widely used in various filed[51–56]. Briefly, TLCC calculates the correlation between two signals by gradually shifting one time series. The correlation peak indicates the time point where two time series are most synchronized. In the case of our measurement, 1 shift (1 lag) represented 3.3 ms (the HS-AFM-LS cycle rate).

Stage drift in HS-AFM-LS can directly be characterized by analyzing the kymograph: X-drift results in a positional shift of the Bcs1 ring in the kymographs, i.e., a lateral shift of the train-track pattern, Y drift leads to a width change of the train-track pattern in the kymograph as the vertical position where the tip crosses the Bcs1 ring would change. X-drift can directly be corrected by comparing each scan line $n$ with the former scan lines and a corrective shift of the entire scan line. Kymographs containing Y-drift were rejected.

## Blue native PAGE gel

NativePAGE™ Novex® 3–12% Bis-Tris Gels were purchased from Thermo-Fisher Scientific. We first prepared 10 tubes with different concentrations of AMP-PNP ranging from 0 to 0.5 mM in buffer containing 20 mM Tris, pH 8.0, 200 mM NaCl, 10 mM MgCl$_2$, 10% glycerol, 0.05% DDM. Then, 6 μg of Bcs1 was added to each tube followed by 30 min incubation at 4 °C. All samples were loaded into native page gel wells and run at 150 V for 60 minutes (at 4 °C), and then run at 250 V for another 60 minutes (at 4 °C). The gel was then fixed using the following steps: (i) Place the gel in fixing solution (40% methanol, 10% acetic acid) and microwave (950–1100 watts) for 45 seconds, (ii) decant the fix solution after shaking for 15 minutes at room

temperature, (iii) add 100 mL 8% acetic acid solution and microwave (950–1100 watts) for 45 seconds, (iv) shake the gel at room temperature until the desired background is obtained. The gel was imaged using a gel imaging systems (Bio-Rad). The images were analyzed in ImageJ. The data was fitted with the following equation[36,57],

$$y = \frac{y_{max}\, C^{n_H}}{EC_{50}{}^{n_H} + C^{n_H}} \qquad (2)$$

where $y_{max}$ is the maximum value of AMP-PNP bound conformational state at saturating ligand concentration, $C$ is the concentration of AMP-PNP, $EC_{50}$ is the concentration at half activation, and $n_H$ is the Hill coefficient.

## Time convolution estimation

In probability theory, the convolution of the individual distributions is the sum of the independent random variables. The probability density function (pdf) of the sum of $n$ random variables results into a gamma distribution with a rate parameter $\lambda$ (Eq. (3))[35],

$$f(x, \lambda, n) = \frac{\lambda^n}{\Gamma(n)}\, x^{n-1} e^{-\lambda x}, x > 0 \qquad (3)$$

where $\Gamma(n)$ is the gamma function.

## Estimation of free energy for opening the TMH of Bcs1

The protein-lipid interaction will distort the membrane mainly due to the hydrophobic mismatch between the lipid and protein TMH hydrophobic region. Considering the structure of the Bcs1 TMH, the deformation free energy is estimated by considering membrane compression and midplane bending. The membrane is modeled as a continuous elastic sheet for these estimations. The compression and bending energy can be expressed as[37,38]

$$G_{(mismatch + radius)} = \pi\kappa_b\left(\frac{\mu}{\lambda}\right)^2\left(1 + \sqrt{2}\frac{r}{\lambda}\right) \qquad (4)$$

where $u(\alpha)$ is half of the mismatch between the hydrophobic region of the protein and the hydrophobic core of the bilayer at angle $\alpha$. $\kappa_b$ is the bilayer bending modulus (~14 $k_BT$), $\lambda$ is the mismatch decay length (~1.1 nm), $r$ is the protein radius (~2 nm for the apo state Bcs1 IMS face, measured from the cryo-EM structure[23]) (Fig. 5e). The tilt energy can be expressed as[37,58]

$$G_{(tilt)} = 2\pi c_0 \kappa \theta r \qquad (5)$$

Where $c_O$ is the spontaneous curvature ($c_0 \approx 1/(10r)$), $\theta$ is membrane slope, $= 90 - \alpha$.

In total, the deformation energy is

$$G_{total} = G_{(mismatch + radius)} + G_{(tilt)} \qquad (6)$$

The total energy change ($\Delta G_{total}$) of the open and closed state as a function of $\alpha$ can be derived (Fig. 5f).

## Reporting summary

Further information on research design is available in the Nature Portfolio Reporting Summary linked to this article.

# Data availability

The manuscript figures, supplementary figures supplementary movies, and source data files contain all data necessary to interpret, verify, and extend the presented work. The raw AFM data is saved as .asd files and can only be opened using proprietary software. The raw data HS-AFM movies and customized codes for data analysis and simulation can be

received from Simon Scheuring (sis2019@med.cornell.edu), upon reasonable request. Source data are provided in this paper.

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

## Acknowledgements

We thank Jeremy Dittman for advice on the subunit coupling time estimation model.

Work in the Scheuring laboratory is supported by grants from the National Institute of Health (NIH), National Center for Complementary and Integrative Health (NCCIH), DP1AT010874, and National Institute of Neurological Disorders and Stroke (NINDS), R01NS110790, and by the Kavli Institute at Cornell. Work in the Xia laboratory was supported by the Intramural Research Program of the NIH, National Cancer Institute, and Center for Cancer Research.

## Author contributions

Y.P. and S.S. designed the experiments; J.Z. and Y.P. expressed and purified the protein; Y.P. performed HS-AFM and HS-AFM-LS experiments; Y.P. performed the HS-AFM and AFM-AFM-LS data analysis; Y.J. performed numerical simulations of the detection probability; D.X. and S.S. supervised all experiments and data analysis. Y.P. and S.S. wrote the paper. All authors edited the manuscript.

## Competing interests

The authors declare no competing interests.
