## [Peer Review File · Nature Communications]

REVIEWER COMMENTS

Reviewer #1 (Remarks to the Author):

In this work authors studied the dynamics of a transmembrane AAA-ATPase (Bcs1) reconstituted in lipid membranes. To characterize the behavior of Bcs1, the authors use high-speed atomic force microscopy combined with a low-dimensional modality (line-scan analysis). Authors showed that the mechanical characteristics captured by HS-AFM are in line with previous data from cryo-EM. Authors further investigated the mechanism by which Bcs1 undergoes conformational change during the ATPase activity.

My major concern is regarding the validity of the line-scan approach for this protein. All supplementary movies show drift. From my understanding line-scan depends on the stability of the system to be able to capture the same line (spatial location) over time. Authors do not elaborate on the drift issues and how they may affect interpretation of the data, and ultimately the validity of their proposed model. Furthermore, the correction of the kymographs for X-drift is not described and raises further questions, e.g. if the drift is 20% how do the authors recapitulate the lost data and how does this affect their interpretation of the data?

Another concern is regarding the resolution during the experiments, authors show in Fig 2d that they are able to distinguish between individual subunits. However, later on the data (e.g. Fig 3) does not show the same high resolution; the high resolution, in my opinion, is critical to obtain mechanistic insight and to support the conclusions drawn in this manuscript.

Reviewer #2 (Remarks to the Author):

The manuscript describes a thorough analysis of the structural dynamics of the Bcs1 ATPase by HS-AFM and HS-AFM-LS. The authors master these techniques remarkably well, probably beyond the state-of-the-art, and the manuscript clearly demonstrates their high suitability for investigation of conformational rearrangements of biological assemblies during their reaction cycle. Bcs1 is AAA+ ATPase unusual in many aspects – it is a planar heptamer in contrast to most of the members of this superfamily that are asymmetric spiral hexamers, and has a transmembrane domain forming a pore which transient opening allows translocation of a folded protein substrate through the inner mitochondrial membrane. The experimental evidence gathered by the authors, combined with the published cryo-EM structures of different nucleotide-bound states of Bcs1, suggest that the mechanism of action of this molecular machine is different from other AAA+ and that all subunits in the ring are always in the same

conformation and undergo only concerted changes. They would (i) bind ATP at the same time, in a highly cooperative manner, resulting in a massive conformational rearrangement of the matrix-facing C-terminal domains that comprise the AAA+ domains, (ii) stay in this conformation for most of the duration of the ATPase cycle and then (iii) rapidly hydrolyse all seven ATPs simultaneously, which would lead to the brief opening of the TMHs, concomitantly with the Pi release, allowing to expell the substrate through the pore to the IMS, in a concerted power stroke.

While I highly recommend this manuscript for publication and think that the proposed mechanism is very plausible, in spite of the thorough analysis of different ATP hydrolysis scenarios in the discussion section, I am still not convinced as to why the authors consider the scenarios i and ii presented in the Supplementary Figure S7b and lines 350-354 as unlikely. The only reason I see in the manuscript itself is the statement in lines 356-357 that «ADP+Pi bound ATP-conformation molecules cannot be found». But what is the proof that they cannot be found? Is this the BN-PAGE gel done with a non hydrolysable ATP analogue AMPPNP? Or is there other, more convincing evidence in the literature? If yes, I would suggest to cite, explain and discuss this evidence explicitly. From what I understood, cryo-EM data is only available on apo, ADP and ATP γ S conformations. Why did the authors then perform the BN-PAGE in AMPPNP and not in ATP γ S as in all the other experiments in the manuscript and in cryo-EM? According to the Expanded Data Figure 1 in Ref. 24, the migration profiles are likely to be similar in both conditions. Do the authors think that one of the analogues (ATP γ S or AMPPNP) is a more faithful representation of the ATP state whereas the other is closer to the ADP-Pi state? I have an impression that if the authors want to really solve the question of the Pi release in this manuscript, then doing AFM and BN-PAGE experiments in ADP with beryllium fluoride or aluminium fluoride to mimick the ADP-Pi state more faithfully would be appropriate. This would however require additional experiments that the authors may not be able to perform during the revision. Alternatively, the corresponding part of the discussion may be reformulated.

Below I provide a few suggestions and list questions answering which may in my opinion, further improve this already excellent and technically very solid manuscript.

1) The abbreviation IMS is used for the first time in line 33 but defined only in line 35. Please correct.

2). The authors refer to the apo conformation as APO, which I find very confusing because too similar to ATP, ADP, etc. I think that in the literature the term apo is normally written in lowercase letters. Please consider editing.

3) Several times in the document, the authors say that generally AAA+ ATPases use a hand-over-hand mechanism. While this model of sequential ATP hydrolysis is indeed often called canonical or orthodox in the AAA+ field, many recent publications, including an AFM study (PMID: 31848341) call it into question. Furthermore, as summarised in this insightful comment (PMID: 32321627), similar structures

can also be interpreted in terms of completely different mechanisms. The current manuscript itself is a perfect illustration of the necessity to combine structural studies with a thorough analysis of protein dynamics by other approaches. Therefore, please consider appropriate rewording.

4) Personally I found the first part of the results section, on the «HS-AFM of Bcs1 in different nucleotide bound states» more difficult to follow than the rest of the manuscript. Tables 1 and 2 would be easier to understand if accompanied by a labelled schematics of the protein structure in different states, showing what exactly is meant by outer diameter, top-ring-diameter, protrusion height, center-to-center distance, etc. In my opinion, figure 1 is not enough.

5) In the explanation of the figure 2d, the authors say that «All HS-AFM imaging data at 1 frame per second image acquisition rate showed that the seven subunits displayed concerted motions for both ATP-binding and -hydrolysis coupled conformational changes». However, before the 7-fold averaging the noisy images corresponding to the individual particles don't allow to claim that all subunits are in the same conformation, and the averaging obviously makes them look identical even although their conformations may have been different. Please consider rewording.

6) Although I generally find the explanations of the AFM-LS analysis very clear, I wonder how the curves would look like if a small fraction of subunits in a small number of heptamers would look like. Would for instance the AFM-LS correlation plots (figure 4c) appear so different that this possibility can be eliminated?

7) Figure 4e: the colors are barely visible, please make them more contrasty.

8) Line 327: «Bcs1 remained in the ATP-conformational for quite long» probably means «ATP-conformational state»

9) Lines 364-367: «First, the nucleotide binding sites are at the interfaces of neighboring subunits, meaning that to form a meaningful binding site both subunits must be in the same conformation, which means in extension that all subunits have to be in the same conformation.» I don't quite follow this logic. Indeed, in all AAA+ ATPases, the ATP binding pochet is completed by the subunit oligomerisation and is located between two neighbouring subunits!

10) Lines 367-371: « Second, the nucleotide binding site in the ATPyS conformation is substantially different from the other conformations... indicative that each conformational state should have an increased affinity for the corresponding nucleotide. » Again, I don't see the logic, why the fact that the

nucleotide binding site in the ATP_γS conformation is different from ADP and apo conformations means that each state has an increased affinity for the corresponding nucleotide ?

11) Legend of the Figure 6. Supplementary Figure S10 doesn't exist, refer to the S7.

Reviewer #3 (Remarks to the Author):

In this manuscript, the authors employ cutting-edge HS-AFM techniques to visualize the conformational changes of the AAA-ATPase, a protein-membrane transporter, with exceptional sub-millisecond time resolution. The most crucial finding of this study is that, contrary to the generally assumed cooperative sequential changes, each protomer of the heptamer undergoes highly coordinated and synchronized conformational transitions during the ATP hydrolysis cycle. The AFM data presented in this study demonstrate exceptional quality, reflecting meticulous experimental techniques. Furthermore, the sophisticated analytical methodology employed, coupled with the comprehensive and insightful discussion, deserves commendation.

I thoroughly enjoyed reading this manuscript and fully support the publication of this study in Nature Communications. However, I believe that the inclusion of supplemental data would further strengthen the conclusions drawn. My comments and suggestions for improvement are outlined below.

1. I have reservations regarding the appropriateness of employing the term "power stroke" in the context of this manuscript. Typically, "power stroke" is used to contrast with the concept of "Brownian ratchet" and describes the direct conversion of ATP hydrolysis energy into mechanical force. In this study, it appears that the authors interpret "power stroke" as a coordinated structural change rather than a sequential "hand-over-hand" motion. To ensure clarity and accuracy, it is suggested that the authors reevaluate their understanding and usage of the term "power stroke" within this context, as there may be potential discrepancies that need to be addressed.

2. Upon examining Figure 1, it is evident that the Bcs1 reconstituted in the lipid membrane is uniformly oriented in the same direction, either towards the IMS side or the matrix side, within the island membrane. Generally, during reconstitution, the orientation would be expected to be randomly determined rather than uniformly aligned. Could the authors explain why this alignment occurred?

3. I wonder why the work lacks an analysis of nucleotide concentration-dependent conformational dynamics and its comparison with ATPase activity turnover. The such analysis seems generally

mandatory for studies of conformational dynamics in ATPases. It is expected that the dwell time of the APO/ADP state shortens depending on the ATP concentration, and the ATP concentration dependence can be discussed whether it follows a Michaelis-Menten or Hill-type behavior. It is expected that the dwell time of the APO/ADP state shortens depending on the ATP concentration, and the ATP concentration dependence can be discussed whether it follows a Michaelis-Menten or Hill-type behavior. Furthermore, it is crucial to compare the quantitative conformational dynamics analyzed from AFM data with the turnover of biochemical ATP hydrolysis (Is it difficult to measure ATP or water due to the presence of lipids?), as this comparison directly relates the observed conformational changes to the chemical reaction of ATP hydrolysis.

4. Related to the above point, it would be valuable to determine if a Hill coefficient similar to the Hill curve measured by SDS can be obtained by examining the number of conformational states as a function of the concentration of AMP-PNP.

5. Although the structure of the APO and ADP states is indistinguishable, if the time constants of the APO/ADP state can be modulated by changing the concentration of phosphate ions in the solution under observation in the presence of ATP or by mixing ADP, ADP-AIF in a ratio, the dynamics of ADP-Pi → ADP → ADP-dissociation may be discussed.

6. The observation of ATP-induced conformational changes on the IMS side is intriguing. On the other hand, it is not obvious whether this conformational dynamic is synchronized with the conformational change on the C-terminal side. Is it possible to analyze the dwell times of the conformational states on the IMS side and compare them to the time constant obtained on the matrix side? Strengthening the relationship between the two states on both sides across the membrane would be valuable. Additionally, if the structural changes on both sides are directly linked, is there a possibility that the conformational dynamics on the matrix side could also be affected by steric hindrance due to contact between the IMS side protrusion and the solid substrate?

7. Regarding the consideration of scenarios discussed in the Discussion section, I agree that scenario i) seems unlikely. However, this could be further clarified by investigating the ATP concentration dependence. I am having difficulty understanding the authors' conclusion for scenario iii) over scenario ii). Is scenario ii) excluded because the dwell time can be fitted with a single exponential function? Also, in the case of scenario iii), is it possible for a protomer to undergo a structural change when only one ATP molecule is bound? If so, would reducing the ATP (or ATP_γS or AMP-PNP) concentration significantly still allow the observation of partial structural changes within the heptamer?

8. There have been several papers analyzing the cooperative conformational changes of ATPase oligomers using HS-AFM, but it appears that none of them have been cited in this manuscript. This lack of citation does not align with scientific fairness and integrity. I kindly request that the authors appropriately cite the relevant literature in their manuscript.

Reviewers' Comments:

REVIEWER #1

Remarks to the Author:

In this work authors studied the dynamics of a transmembrane AAA-ATPase (Bcs1) reconstituted in lipid membranes. To characterize the behavior of Bcs1, the authors use high-speed atomic force microscopy combined with a low-dimensional modality (line-scan analysis). Authors showed that the mechanical characteristics captured by HS-AFM are in line with previous data from cryo-EM. Authors further investigated the mechanism by which Bcs1 undergoes conformational change during the ATPase activity.

General: We thank the reviewer for their assessment and critique of our work. Indeed, while cryo-EM solved the structural snapshots of the AAA-ATPase Bcs1, it was our objective using HS-AFM to monitor the conformational changes, determine the kinetics, and to decipher the mechanism, i.e., whether the conformational transition was sequential or concerted.

Comment 1: My major concern is regarding the validity of the line-scan approach for this protein. All supplementary movies show drift. From my understanding line-scan depends on the stability of the system to be able to capture the same line (spatial location) over time. Authors do not elaborate on the drift issues and how they may affect interpretation of the data, and ultimately the validity of their proposed model. Furthermore, the correction of the kymographs for X-drift is not described and raises further questions, e.g. if the drift is 20% how do the authors recapitulate the lost data and how does this affect their interpretation of the data?

Response 1: The reviewer is concerned about the stage drift during HS-AFM-LS acquisition. We agree with the reviewer that drift is a general concern with HS-AFM-LS studies. The previous studies (Heath, G. R. & Scheuring, S. *Nat. Commun.* **9**, 4983 (2018), Matin, TR. *Nat. Commun.* **11**, 5016 (2020).), reported that the stage drift varies from ~1.0 to 0.02 nm/s, depending on experimental duration and thermal equilibration. It is worth noting that we consistently allow the system to equilibrate and reach a state of equilibrium prior to conducting HS-AFM-LS. This crucial step minimizes drift and ensures optimal accuracy and reliability in our measurements. In addition, as mentioned in the Methods/Data analysis, the stage drift in HS-AFM-LS can easily be characterized by analyzing the kymograph (Figure R1): X-drift results in a positional shift of the Bcs1 ring in the kymographs, i.e. lateral shift of the train-track pattern, Y drift leads to a width change of the train-track pattern in the kymograph as the vertical position where the tip crosses the Bcs1 ring would change. X-drift can easily and directly be corrected for. Kymographs containing Y-drift were rejected – as soon as the train-tracks narrowed we know that we are not crossing the ring anymore in the central region. Regarding X-drift, the reviewer is concerned about the correction of the kymographs and data loss. X-drift causes a positional shift of Bcs1 in the kymographs (Figure R1), but it is important to note that no data is lost during the alignment process as all pixels in the entire LS scan line are shifted together. To address this issue, we employed an algorithm in ImageJ for aligning the kymographs: Comparing iteratively scan line n with prior scan lines. In revision, we detail about drift in HS-AFM-LS and its correction in the Methods section.

Figure R1 | Simulation of X (left) and Y (right) stage drifts and the outcome in Bcs1 HS-AFM-LS kymographs.

Comment 2: Another concern is regarding the resolution during the experiments, authors show in Fig 2d that they are able to distinguish between individual subunits. However, later on the data (e.g. Fig 3) does not show the same high resolution; the high resolution, in my opinion, is critical to obtain mechanistic insight and to support the conclusions drawn in this manuscript.

Response 2: We of course agree with the reviewer that the resolution is important. However, it is important to note that Bcs1 displayed a substantially different conformation, the matrix cavity size is reduced by nearly 70% and the height of the nucleotide binding domains decreased by $\sim 7 \text{ \AA}$, upon ATP γ S (non-hydrolyzable ATP) binding. In addition, Bcs1 appears ring structure in apo/ADP conformational state due to its large central cavity. All these differences between the APO/ADP and the ATP conformations make Bcs1 an ideal system for structure-dynamics studies of the ATPase cycle by HS-AFM. Indeed, to assess the conformational dynamics and thus to characterize the enzymatic cycle, the height difference alone would be entirely sufficient, and the sensitivity in the z-axis (largely independent of the x,y resolution) of HS-AFM gives a tremendously strong signal, see Figure R2 and all kymographs in the main manuscript.

In our HS-AFM-LS experiments, the tip scans across the central pore of the molecule. Therefore, the HS-AFM-LS kymographs have train-track appearance as expected when the central cavity in our experiments is clearly resolved in the apo/ADP conformation (Figure 3a,b). In the ATP (ATP γ S) conformation, it may not always be feasible to differentiate between neighboring subunits in the images, but the peripheral pixels in the train-tracks report unambiguously about the height of the two peripheral subunits in the ring, and therefore our thorough analysis revealed synchronous conformational changes on the left and right sides of the molecule (Figure R2). This is strong evidence for the concerted motion (or faster coupling than the $270\mu\text{s}$ that is needed for the tip to cross the ring) of the subunits. Thus, the ring structure of Bcs1 and the substantial conformational change especially in height (7\AA) between the APO/ADP and ATP conformational states, the design of the HS-AFM-LS and the thorough analysis (Figure 4a-f) allowed to draw the conclusion that Bcs1 had a highly concerted conformational change.

Figure R2 | Schematic showing the design of our HS-AFM-LS.

REVIEWER #2

Remarks to the Author:

The manuscript describes a thorough analysis of the structural dynamics of the Bcs1 ATPase by HS-AFM and HS-AFM-LS. The authors master these techniques remarkably well, probably beyond the state-of-the-art, and the manuscript clearly demonstrates their high suitability for investigation of conformational rearrangements of biological assemblies during their reaction cycle. Bcs1 is AAA+ ATPase unusual in many aspects – it is a planar heptamer in contrast to most of the members of this superfamily that are asymmetric spiral hexamers, and has a transmembrane domain forming a pore which transient opening allows translocation of a folded protein substrate through the inner mitochondrial membrane. The experimental evidence gathered by the authors, combined with the published cryo-EM structures of different nucleotide-bound states of Bcs1, suggest that the mechanism of action of this molecular machine is different from other AAA+ and that all subunits in the ring are always in the same conformation and undergo only concerted changes. They would (i) bind ATP at the same time, in a highly cooperative manner, resulting in a massive conformational rearrangement of the matrix-facing C-terminal domains that comprise the AAA+ domains, (ii) stay in this conformation for most of the duration of the ATPase cycle and then (iii) rapidly hydrolyse all seven ATPs simultaneously, which would lead to the brief opening of the TMHs, concomitantly with the Pi release, allowing to expell the substrate through the pore to the IMS, in a concerted power stroke.

General: We thank the reviewer for their overall positive assessment of our work and for noting the “probably beyond the state-of-the-art” quality of our HS-AFM data.

While I highly recommend this manuscript for publication and think that the proposed mechanism is very plausible, in spite of the thorough analysis of different ATP hydrolysis scenarios in the discussion section, I am still not convinced as to why the authors consider the scenarios i and ii presented in the Supplementary Figure S7b and lines 350-354 as unlikely. The only reason I see in the manuscript itself is the statement in lines 356-357 that «ADP+Pi bound ATP-conformation molecules cannot be found». But what is the proof that they cannot be found? Is this the BN-PAGE gel done with a non hydrolysable ATP analogue AMPPNP? Or is there other, more convincing evidence in the literature? If yes, I would suggest to cite, explain and discuss this evidence explicitly. From what I understood, cryo-EM data is only available on apo, ADP and ATP γ S conformations. Why did the authors then perform the BN-PAGE in AMPPNP and not in ATP γ S as in all the other experiments in the manuscript and in cryo-EM? According to the Expanded Data Figure 1 in Ref. 24, the migration profiles are likely to be similar in both conditions. Do the authors think that one of the analogues (ATP γ S or AMPPNP) is a more faithful representation of the ATP state whereas the other is closer to the ADP-Pi state? I have an impression that if the authors want to really solve the question of the Pi release in this manuscript, then doing AFM and BN-PAGE experiments in ADP with beryllium fluoride or aluminium fluoride to mimic the ADP-Pi state more faithfully would be appropriate. This would however require additional experiments that the authors may not be able to perform during the revision. Alternatively, the corresponding part of the discussion may be reformulated.

Response: We thank the reviewer for their insightful and constructive comment. Indeed, while HS-AFM is highly informative about the conformational states of the subunits in the ring, it is much more difficult to assign the nucleotide occupancy state to the series of events.

Regarding the rationale for using AMP-PNP (and not ATP γ S) in the BN-PAGE gel, we do not think that one of the analogues (ATP γ S or AMPPNP) is a more faithful representation of the ATP state whereas the other is closer to the ADP-Pi state. We think that both represent the ATP conformational state as reported in Ref 24 (Di Xia et.al., Nat Struct Mol Biol. 2020 Feb;27(2):202-209). We used AMP-PNP, because (for reasons that we do not understand) the bands in the BN-PAGE gel were neater than when doing the same experiments in ATP γ S.

For revision, we performed several additional experiments aiming at answering the reviewer's questions:

First, we performed imaging of Bcs1 as a function of AMP-PNP concentration. The AMP-PNP conformational state shares the essential topography signatures with the ATP γ S and ATP conformational states (new Figure 4i). Thus we titrated AMP-PNP into the HS-AFM fluid cell and counted the apo and AMP-PNP state molecules, which allowed us to determine an EC50 and a Hill coefficient for the AMP-PNP binding.

Second, we performed a series of additional HS-AFM-LS experiments and measured the dwell times of ATP and apo/ADP conformational states at different ATP concentration. We found that the ATP state was ATP concentration independent, while the apo/ADP state shortened with ATP concentration, both as expected (new panels Figure 3e-h).

Third, we also measured the binding affinity of ADP or phosphate to Bcs1 (new Supplementary Figure 10).

In light of the new data, we rephrased our discussion (lines 370-381).

Below I provide a few suggestions and list questions answering which may in my opinion, further improve this already excellent and technically very solid manuscript.

Response: We thank the reviewer for the constructive suggestions. We have amended our manuscript accordingly and hope to address all the reviewer's comments satisfactorily.

Comment 1: The abbreviation IMS is used for the first time in line 33 but defined only in line 35. Please correct.

Response 1: We thank the reviewer for pointing out this. In revision, this has been corrected.

Comment 2. The authors refer to the apo conformation as APO, which I find very confusing because too similar to ATP, ADP, etc. I think that in the literature the term apo is normally written in lowercase letters. Please consider editing.

Response 2: We agree. To avoid confusion, we changed 'APO' to 'apo' throughout the revised manuscript.

Comment 3. Several times in the document, the authors say that generally AAA+ ATPases use a hand-over-hand mechanism. While this model of sequential ATP hydrolysis is indeed often called canonical or orthodox in the AAA+ field, many recent publications, including an AFM study (PMID: 31848341) call it into question. Furthermore, as summarised in this insightful comment (PMID: 32321627), similar structures can also be interpreted in terms of completely different mechanisms. The current manuscript itself is a perfect illustration of the necessity to combine structural studies with a thorough analysis of protein dynamics by other approaches. Therefore, please consider appropriate rewording.

Response 3: We thank the reviewer for these constructive suggestions. In revision we reference to the indicated papers and have amended the discussion in lines 385-395.

Comment 4. Personally I found the first part of the results section, on the «HS-AFM of Bcs1 in different nucleotide bound states» more difficult to follow than the rest of the manuscript. Tables 1 and 2 would be easier to understand if accompanied by a labelled schematics of the protein structure in different states, showing what exactly is meant by outer diameter, top-ring-diameter, protrusion height, center-to-center distance, etc. In my opinion, figure 1 is not enough.

Response 4: We regret that the reviewer thought that the outer diameter, top-ring-diameter, protrusion height, center-to-center distance is not well illustrated until Figure S1 and S2. In the revised manuscript, we combined Figure 1 and the original Figure S2. Additionally, we have improved the clarity of Figure 1 and Figure S1, and in consideration of the visual presentation, specifically grouping the illustrations and data analysis (histograms) together, we have chosen to retain Figure S1 in the Supplementary data.

Comment 5. In the explanation of the figure 2d, the authors say that «All HS-AFM imaging data at 1 frame per second image acquisition rate showed that the seven subunits displayed concerted motions for both ATP-binding and -hydrolysis coupled conformational changes». However, before the 7-fold averaging the noisy images corresponding to the individual particles don't allow to claim that all subunits are in the same conformation, and the averaging obviously makes them look identical even although their conformations may have been different. Please consider rewording.

Response 5: The reviewer is concerned about the imaging resolution. While we agree with the reviewer that our phrasing "All HS-AFM imaging data at 1 frame per second image acquisition rate showed that the seven subunits displayed" is not precise and needed rephrasing (lines 159-161.), we disagree with the reviewer on the scientific state assignment. The ATP state images in Fig 2d clearly reveal the 7 subunits without symmetrizing, but we agree

that the ADP/apo state and other complexes in panel a) not. Indeed, seven peripheral protrusions can be identified for the ATP conformation (79 and 81 s) (Figure 2d, Figure R3). Most importantly, Bcs1 displayed a substantial height change, $\sim 7 \text{ \AA}$, between the ATP and the ADP/apo state, which allows unambiguous assignment of all positions on the rings to the ATP or the ADP/apo state. When we extract the height distribution histograms of all pixels in the area that corresponding to the protrusions (Figure R3, white dash cycles). We found that the peak value and distributions for all histograms are almost identical, 6.2-6.4 nm, in the ATP conformation (the seven subunits can also be identified if one observes them carefully; see Figure 2d), while the peak value and distributions are again rather consistent, 6.8-7.0 nm, in the ADP/apo conformation, and very different between states. All these analyses are in favor of our claim that all subunits are in the same conformation. Accordingly, we added new Figures S2 in the revised our manuscript.

Figure R3 |. Height distribution histogram of all pixels in the white cycle in the left image.

Comment 6. Although I generally find the explanations of the AFM-LS analysis very clear, I wonder how the curves would look like if a small fraction of subunits in a small number of heptamers would look like. Would for instance the AFM-LS correlation plots (figure 4c) appear so different that this possibility can be eliminated?

Response 6: In our time-lag cross-correlation analysis, the two height/time profiles of the peripheral subunits (the only subunits contoured in HS-AFM-LS) are shifted with respect to each other and the cross-correlation for each shift calculated (Figure 4b; Figure R4). If the conformational step was 1 scan line earlier or later in one of the two peripheral subunits, then the cross-correlation signal would give an unambiguous signal at -1 or +1 time lag in the cross-correlation plots (see Figure 4b, right). Figure 4c is a false-color scale representation of all the lag-time cross-correlation plots (as in Figure 4c, right). If a subunit had a -1 or +1 time lag shift in the height-time traces, the most red pixel in Figure 4c, the time lag with the highest cross-correlation, would be shifted by one pixel on the Y-axis (lag time), but this was never observed.

Figure R4 | Illustrations of time lagged cross correlation analysis
(<http://robosub.eecs.wsu.edu/wiki/ee/hydrophones/start>).

Comment 7. Figure 4e: the colors are barely visible, please make them more contrasty.

Response 7: We thank the reviewer for pointing this. In the revision, we adapted the contrast between the lines.

Comment 8. Line 327: «Bcs1 remained in the ATP-conformational for quite long» probably means «ATP-conformational state»

Response 8: Yes, we corrected this.

Comment 9. Lines 364-367: «First, the nucleotide binding sites are at the interfaces of neighboring subunits, meaning that to form a meaningful binding site both subunits must be in the same conformation, which means in extension that all subunits have to be in the same conformation.» I don't quite follow this logic. Indeed, in all AAA+ ATPases, the ATP binding pocket is completed by the subunit oligomerisation and is located between two neighbouring subunits!

Response 9: The reviewer is correct, with regard to the location of the binding sites. We wanted to convey the message that in Bcs1, the structures are ATP γ S- or ADP-bound or apo in the conformation where all subunits are in the same conformation, and therefore it appears that in Bcs1, nucleotide-specific binding pockets are located between subunits in the same conformation in contrast to other AAA+ ATPases. Anyway, to avoid confusion, this sentence was removed.

Comment 10. Lines 367-371: « Second, the nucleotide binding site in the ATP γ S conformation is substantially different from the other conformations... indicative that each conformational state should have an increased affinity for the corresponding nucleotide. » Again, I don't see the logic, why the fact that the nucleotide binding site in the ATP γ S conformation is different from ADP and apo conformations means that each state has an increased affinity for the corresponding nucleotide ?

Response 10: It seems logical to us that a binding site that fits a specific molecules size and shape has likely a better affinity than a binding site that is not fitting the molecule. It appears that the ADP conformational state binding site would be too small to host an ATP molecule, thus one would expect ATP to have a lower affinity to the ADP/apo conformation. If this was not the case, then one would have ADP/apo conformation with ATP bound and the complex would remain stable in this configuration (and a structure could be solved), clearly the ATP conformation with ATP bound is the lower energy state.

Comment 11. Legend of the Figure 6. Supplementary Figure S10 doesn't exist, refer to the S7.

Response 11: We thank the reviewer for pointing this. We amended accordingly.

REVIEWER #3

Remarks to the Author:

In this manuscript, the authors employ cutting-edge HS-AFM techniques to visualize the conformational changes of the AAA-ATPase, a protein-membrane transporter, with exceptional sub-millisecond time resolution. The most crucial finding of this study is that, contrary to the generally assumed cooperative sequential changes, each protomer of the heptamer undergoes highly coordinated and synchronized conformational transitions during the ATP hydrolysis cycle. The AFM data presented in this study demonstrate exceptional quality, reflecting meticulous experimental techniques. Furthermore, the sophisticated analytical methodology employed, coupled with the comprehensive and insightful discussion, deserves commendation.

I thoroughly enjoyed reading this manuscript and fully support the publication of this study in Nature Communications. However, I believe that the inclusion of supplemental data would further strengthen the conclusions drawn. My comments and suggestions for improvement are outlined below.

Response: We thank the reviewer for their overall positive assessment of our work and for pointing out the high quality of the data. We have amended our manuscript according to reviewer's recommendations and hope to address all the reviewer's comments satisfactorily.

Comment 1. I have reservations regarding the appropriateness of employing the term "power stroke" in the context of this manuscript. Typically, "power stroke" is used to contrast with the concept of "Brownian ratchet" and describes the direct conversion of ATP hydrolysis energy into mechanical force. In this study, it appears that the authors interpret "power stroke" as a coordinated structural change rather than a sequential "hand-over-hand" motion. To ensure clarity and accuracy, it is suggested that the authors reevaluate their understanding and usage of the term "power stroke" within this context, as there may be potential discrepancies that need to be addressed.

Response 1: We agree with the reviewer that the usage of the word power stroke is strongly coined, we removed the mention throughout the manuscript and reworded the corresponding parts in Results and Discussion.

Comment 2. Upon examining Figure 1, it is evident that the Bcs1 reconstituted in the lipid membrane is uniformly oriented in the same direction, either towards the IMS side or the matrix side, within the island membrane. Generally, during reconstitution, the orientation would be expected to be randomly determined rather than uniformly aligned. Could the authors explain why this alignment occurred?

Response 2: This is an interesting observation that we did not directly address in our manuscript. In general, we found the following tendencies when growing 2D crystals or densely packed vesicles: When reconstituting small membrane proteins without substantial hydrophilic domains that stick out of the membrane, the crystal contacts are dominated by the hydrophobic regions, and the reconstitutions often comprise up-and-down oriented proteins or randomly oriented densely packed proteins (e.g., aquaporins, see Jiang et al, Nature Communications 2023). In contrast, for the membrane proteins that comprise large hydrophilic extra-membranous domain, the hydrophilic interactions may also play an important role in 2D crystallization or dense packing (eg. GLIC, see Ruan et al, PNAS 2018). Despite Bcs1 being a heptamer, which theoretically cannot form ordered 2D crystals, Bcs1 possesses large extramembrane domains (Figure 1a). In our reconstitutions, Bcs1 molecules are densely packed, indicating that hydrophilic interactions of the extra membranous domains likely provide a strong bias to the orientation of these molecules during the reconstitution process (lines 90-94).

Comment 3. I wonder why the work lacks an analysis of nucleotide concentration-dependent conformational dynamics and its comparison with ATPase activity turnover. The such analysis seems generally mandatory for studies of conformational dynamics in ATPases. It is expected that the dwell time of the APO/ADP state shortens depending on the ATP concentration, and the ATP concentration dependence can be discussed whether it follows a Michaelis-Menten or Hill-type behavior. It is expected that the dwell time of the APO/ADP state shortens depending on the ATP concentration, and the ATP concentration dependence can be discussed whether it follows a Michaelis-Menten or Hill-type behavior. Furthermore, it is crucial to compare the quantitative conformational dynamics analyzed from AFM data with the turnover of biochemical ATP hydrolysis (Is it difficult to measure ATP or water due to the presence of lipids?), as this comparison directly relates the observed conformational changes to the chemical reaction of ATP hydrolysis.

Response 3: We thank the reviewer for this constructive comment. In the original submission, we were particularly interested in answering the question whether Bcs1 worked in a concerted or sequential manner. Following the reviewer's recommendation, we now present ATP concentration-dependent experiments (new panels Figure 3e-h). We found that the dwell time of the Apo/ADP conformational state shortens with ATP concentration, as reviewer pointed out, while the ATP conformational state was of constant duration, as expected. The dwell time of the Apo/ADP conformational state shortens until it reaches a plateau of ~0.32 s, at 5 μ M ATP concentration or higher. The dwell time of ATP conformational state is ~5.6 s, independent of ATP concentrations. ATP hydrolysis is thus the rate-limiting steps in the enzymatic reaction under saturating ATP concentration.

Comment 4. Related to the above point, it would be valuable to determine if a Hill coefficient similar to the Hill curve measured by SDS can be obtained by examining the number of conformational states as a function of the concentration of AMP-PNP.

Response 4: We greatly appreciate the reviewer's valuable suggestion. In response, we performed imaging of Bcs1 as a function of AMP-PNP concentration and subsequently calculated the probability distribution of AMP-PNP conformational states. The resulting data was fitted using the Hill equation, leading to the generation of a new Figure 4i. Encouragingly, our analysis revealed that the EC_{50} and Hill coefficient values obtained from the high-speed atomic force microscopy (HS-AFM) data matched those determined through BN-PAGE gel analysis.

5. Although the structure of the APO and ADP states is indistinguishable, if the time constants of the APO/ADP state can be modulated by changing the concentration of phosphate ions in the solution under observation in the

presence of ATP or by mixing ADP, ADP-AIF in a ratio, the dynamics of ADP-Pi → ADP→ADP-dissociation may be discussed.

Response 5: We thank the reviewer for this suggestion. We measured the dwell time of the Apo/ADP state in 10 μM ATP solutions supplemented with 200 μM phosphate ions. We did not observe changes in the dwell time of the Apo/ADP conformations, when comparing this data (new Figure S9) to the results obtained in the absence of additional phosphate. To get more information, we measured the binding affinity between Bcs1 and phosphate ions in bulk by performing microscale thermophoresis (MST) experiments (new Figure S10) and found that the affinity of phosphate ions to Bcs1 (KD value ~3.1 mM). Additionally, we determined the affinity of ADP to Bcs1 (KD value ~0.8 μM). Both were lower than the Km for ATP, 0.2 μM. Thus, phosphate should leave the complex immediately, and given that the turnover experiments are performed in high ATP and low ADP concentrations (as in the mitochondria), ADP should also leave the complex swiftly, and what we call apo/ADP state is likely the apo state. These findings are now discussed in more detail in our revised manuscript (lines 364-367).

Comment 6. The observation of ATP-induced conformational changes on the IMS side is intriguing. On the other hand, it is not obvious whether this conformational dynamic is synchronized with the conformational change on the C-terminal side. Is it possible to analyze the dwell times of the conformation states on the IMS side and compare them to the time constant obtained on the matrix side? Strengthening the relationship between the two states on both sides across the membrane would be valuable. Additionally, If the structural changes on both sides are directly linked, is there a possibility that the conformational dynamics on the matrix side could also be affected by steric hindrance due to contact between the IMS side protrusion and the solid substrate?

Response 6: We thank the reviewer for the comment. Indeed, we present the single molecule Bcs1 IMS face volume/time traces in ATP conditions (Figure 5b, *left*), in which we found spikes in the volume/time traces only last 1s which is our imaging time resolution. Thus the dwell time of the IMS side open state should be 1 second or shorter while the ATP state on the matrix side has a dwell time of ~5.6 seconds. We have considered performing HS-AFM-LS experiments on the IMS side, but noted that due to the small conformational change of the IMS, we needed to analyze the conformational changes as a function of time by measuring the protrusion volume instead of height or width alone in our HS-AFM images to achieve a better signal and more accurate measurements. Thus, due to the small conformational change and low-dimensional modality of HS-AFM-LS, accurately capturing these measurements using HS-AFM-LS poses a significant challenge. Instead, we estimated the probability of the molecules in the larger-volume population (Figure 5b, *right*), and found that the larger-volume population on the IMS face was rare, ~5%, while the matrix face is ~90% of the cycle in the ATP-bound conformation (~5.6s of the total cycle of ~5.9s). Thus, we can exclude that the molecules within the larger-volume population on the IMS face represent directly the other face of the ATP-bound conformation.

Considering the reviewer's comment regarding the potential impact of Bcs1 membrane adsorption on the mica sample support on conformational dynamics. It is indeed a common concern in AFM studies involving membrane proteins that the sample must be deposited on a support. However, it should be noted that the physisorption process, in which no chemical or biological bond is formed between the sample and the surface, implies that a buffer layer becomes entrapped between the supported lipid bilayer and the mica surface. The presence of this buffer layer helps to maintain the structural and functional properties of the membrane proteins (Andreas Engel et al., *Biophysical Journal* 73, 1997, 1633-1644). In addition, in our function analysis, we found that the dwell time of Apo/ADP conformational states is ATP concentration dependent. While the dwell time of ATP conformation state is independent of ATP concentrations. The ATPase activity measured at the single molecule level (new Figure S3h) is comparable with that derived from the bulk experiments (lines 211-213) (Di Xia et.al., *Nat Struct Mol Biol.* 2020 Feb;27(2):202-209). This suggests that Bcs1 functions in a physiological manner. Thus, even if such protein-support interactions would occur, there should be only minor steric hindrance for conformational changes on the matrix side of the protein.

Comment 7. Regarding the consideration of scenarios discussed in the Discussion section, I agree that scenario i) seems unlikely. However, this could be further clarified by investigating the ATP concentration dependence. I am having difficulty understanding the authors' conclusion for scenario iii) over scenario ii). Is scenario ii) excluded because the dwell time can be fitted with a single exponential function Also, in the case of scenario iii), is it possible for a protomer to undergo a structural change when only one ATP molecule is bound? If so, would reducing the ATP (or ATPγS or AMP-PNP) concentration significantly still allow the observation of partial structural changes within the heptamer?

Response 7: Regarding the various nucleotide state scenarios in the Discussion section: After thoughtful consideration, we think that the high Hill coefficient, $n_H = \sim 6$, to reach the AMP-PNP conformation, in possible agreement with all three scenarios. Scenarios (i) and (iii) suggest binding cooperativity, while scenario (ii) rather suggests multiplicity of binding for the conformational change. However, from a structural perspective, analyzing the nucleotide binding sites in the apo, ADP and ATP γ S conformations, we note that the nucleotide binding site in the ATP γ S conformation is substantially different from the other conformations (RMSDs: ATP γ S vs apo: 10.1 Å, ATP γ S vs ADP: 10.5 Å, ADP vs apo: 1.8 Å) indicative that the ATP γ S conformation has an increased affinity for ATP, and therefore we propose that scenario (iii) is the most plausible, because the other scenarios imply ATP binding to the apo/ADP conformation. Further work is needed to test this hypothesis. (Lines, 353-368). Regarding the question whether a protomer can undergo a structural change with just one bound ATP molecule. As shown in Figure 4, we attained 270 μ s temporal resolution, which is around ~ 1200 times faster than the apo/ADP conformation state dwell time (320 ms). If there were partial structural changes within the heptamer, we calculated the probability of detecting asymmetric train-track-like appearances in our kymographs-but we never did, which is why we exclude the probability of a protomer undergoing a structural change when only one ATP molecule is bound.

Comment 8. There have been several papers analyzing the cooperative conformational changes of ATPase oligomers using HS-AFM, but it appears that none of them have been cited in this manuscript. This lack of citation does not align with scientific fairness and integrity. I kindly request that the authors appropriately cite the relevant literature in their manuscript.

Response 8: We thank the reviewer for this suggestion. In revision, we cited the relevant HS-AFM work about ATPase oligomers (refs 25,26,27,28).

REVIEWERS' COMMENTS

Reviewer #2 (Remarks to the Author):

I thank the authors for having thoroughly addressed all my concerns. I have no further substantial comments or request and congratulate the authors for this excellent work.

Reviewer #3 (Remarks to the Author):

The authors respond appropriately to the reviewers' comments, and further experimental data and analysis lead to more convincing and firm conclusions. Thus I am pleased to endorse the manuscript for publication.

Reviewers' Comments:

REVIEWER #2

Remarks to the Author:

I thank the authors for having thoroughly addressed all my concerns. I have no further substantial comments or request and congratulate the authors for this excellent work.

We thank the reviewer for their initial comments and constructive suggestions that helped us improve our work. We also thank the reviewer for their kind words regarding our final version.

REVIEWER #3

Remarks to the Author:

The authors respond appropriately to the reviewers' comments, and further experimental data and analysis lead to more convincing and firm conclusions. Thus I am pleased to endorse the manuscript for publication.

We thank the reviewer for their initial comments and constructive suggestions that helped us improve our work. We also thank the reviewer for their kind words regarding our final version.